# UltraHR-100K: Enhancing UHR Image Synthesis with A Large-Scale High-Quality Dataset

**Chen Zhao[1,*], En Ci[1,*], Yunzhe Xu[1,*], Tiehan Fan[1], Shanyan Guan[2],**

**Yanhao Ge[2], Jian Yang[1], Ying Tai[1,†]**

[1] State Key Laboratory of Novel Software Technology, Nanjing University, China
[2] vivo Mobile Communication Co., Ltd., China

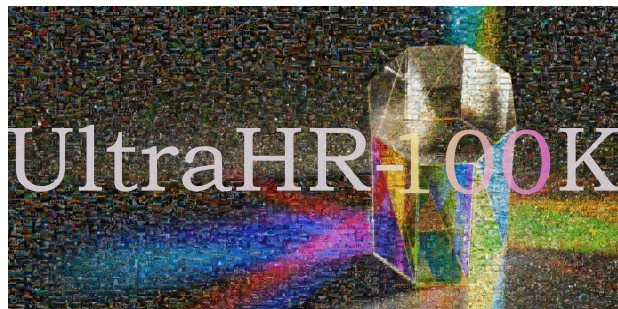 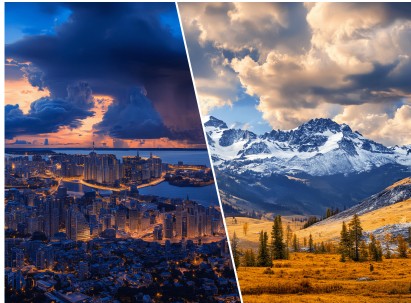

Figure 1: Our UltraHR-100K (**left**) is a large-scale high-quality dataset for ultra-high-resolution (UHR) image synthesis, featuring a diverse range of categories. Utilizing this dataset enables the generation of high-fidelity UHR images (**right**).

## Abstract

Ultra-high-resolution (UHR) text-to-image (T2I) generation has seen notable progress. However, two key challenges remain : 1) the absence of a large-scale high-quality UHR T2I dataset, and (2) the neglect of tailored training strategies for fine-grained detail synthesis in UHR scenarios. To tackle the first challenge, we introduce **UltraHR-100K**, a high-quality dataset of 100K UHR images with rich captions, offering diverse content and strong visual fidelity. Each image exceeds 3K resolution and is rigorously curated based on detail richness, content complexity, and aesthetic quality. To tackle the second challenge, we propose a frequency-aware post-training method that enhances fine-detail generation in T2I diffusion models. Specifically, we design (i) *Detail-Oriented Timestep Sampling (DOTS)* to focus learning on detail-critical denoising steps, and (ii) *Soft-Weighting Frequency Regularization (SWFR)*, which leverages Discrete Fourier Transform (DFT) to softly constrain frequency components, encouraging high-frequency detail preservation. Extensive experiments on our proposed UltraHR-eval4K benchmarks demonstrate that our approach significantly improves the fine-grained detail quality and overall fidelity of UHR image generation. The code is available at here.

---

*Equal Contribution.
†Correspondence to: Ying Tai.

39th Conference on Neural Information Processing Systems (NeurIPS 2025).

# 1 Introduction

Recent advances in text-to-image (T2I) diffusion models have greatly improved image quality and controllability [1, 2, 3, 4, 5, 6, 7, 8, 9, 10, 11]. However, most existing models are constrained to fixed resolutions (typically 1024×1024), and exhibit noticeable quality degradation and structural artifacts when directly scaled to ultra-high-resolution (UHR) image generation [12, 13, 14, 15, 16, 17, 18, 19]. This limitation poses a significant barrier for real-world applications that demand fine-grained detail and high visual fidelity, such as digital art, virtual content creation, and commercial design.

Existing solutions to face this challenge can be grouped into two main paradigms: training-free [13, 14, 20, 21, 22, 23, 24, 25, 26] and training-based methods [12, 15, 16, 17, 27]. Training-free methods attempt to generate UHR images by modifying network architectures [20, 22, 23] or by adjusting inference schemes [14, 21]. However, these techniques exhibit excessive smoothing, produce implausible details, and incur prolonged inference times—limitations that severely hinder their practical deployment [12]. Fundamentally, training-free methods depend on pre-trained T2I models [2, 3, 5, 7] that were not exposed to UHR data during training, and consequently *lack the inherent capacity to render the fine-grained, photorealistic details essential* that real-world UHR image synthesis requires.

Recently, training-based models for UHR image generation have shown promising results [15, 16, 17]. However, they still face two critical challenges: 1) *The absence of a open-source, large-scale high-quality UHR T2I dataset.* High-fidelity UHR image collection is burdensome due to the scarcity of suitable data. Although Aesthetic-4K [16] introduced the first open-source UHR T2I dataset, it remains limited in both scale (approximately 10K images) and quality (the lack of a rigorous selection criterion), constraining its generalizability and high-quality generation capabilities in real-world scenarios. Consequently, constructing a open-source, large-scale high-quality UHR T2I dataset represents both a significant challenge and a critical necessity. 2) *The neglect of tailored training strategies for UHR fine-grained detail synthesis.* Existing models primarily focus on training efficiency to fine-tune pre-trained T2I models [15, 17], overlooking the high-fidelity detail synthesis. Large-scale pre-training equips T2I models with strong semantic planning abilities, but they struggle to synthesize fine-grained details in the UHR setting [2, 3, 5, 7]. Thus, a detail-oriented training strategy is essential for achieving high-quality UHR image synthesis.

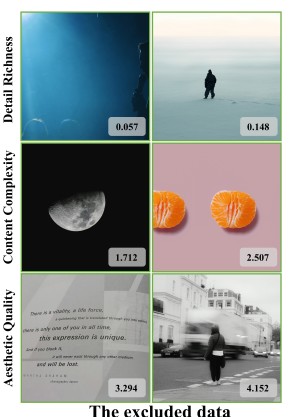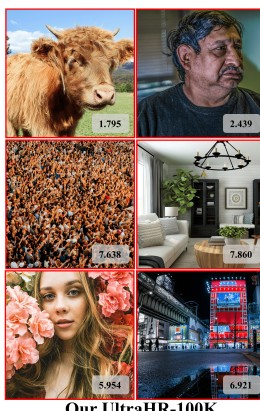

Figure 2: We perform a rigorous selection of UltraHR-100K by evaluating all collected images across three key dimensions: detail richness, content complexity, and aesthetic quality. **Left:** We present representative low-quality (bad case) examples for each dimension along with their corresponding scores, highlighting the necessity of such filtering. **Right:** In contrast, our UltraHR-100K exhibit superior texture details, semantic complexity, and aesthetic appeal.

**Large-Scale High-Quality Dataset for Tackling Challenge 1:** We construct UltraHR-100K, a large-scale high-quality UHR T2I dataset consisting of 100K UHR images paired with rich textual descriptions. As illustrated in Figure 1, UltraHR-100K offers the following key advantages: 1) *Scale and Diversity*: Compared to recent publicly available Aesthetic-4K [16], our UltraHR-100K is approximately 10× larger, featuring 100K images spanning a broad spectrum of categories and visual concepts. 2) *Higher Quality*: All images in UltraHR-100K are rigorously selected from three key dimensions: detail richness, content complexity, and aesthetic quality. Notably, the minimum resolution across the proposed dataset exceeds 3K (average of width and height), ensuring high-resolution content, as shown in Figure 3. 3) *Fine-Grained Captions*: To provide detailed textual annotations for each image, we leverage Gemini 2.0 [28], a powerful commercial vision-language model (VLM), to generate high-quality captions. As shown in Figure 4, our captions are significantly more detailed and semantically rich compared to those in the Aesthetic-4K [16].

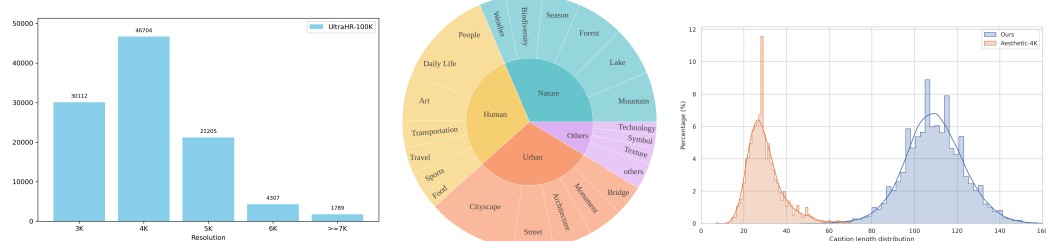

Figure 3: **Left:** Resolution distribution of our UltraHR-100K. All images have a minimum resolution of 3K, defined as the average of height and width exceeding 3000 pixels. **Middle:** Image categories across our dataset. *The proportion of each category mirrors its distribution in our dataset.* **Right:** Caption length distribution. Compared to the recent Aesthetic-4K[16], our captions are significantly longer, providing richer semantic supervision.

Table 1: Overview of our data processing pipeline. The first stage involves large-scale data collection and preliminary filtering to ensure a baseline level of visual quality. The second stage performs three parallel filtering procedures. The final high-quality dataset is obtained by taking the intersection of these subsets. We further employ a strong VLM (Gemini 2.0) to annotate each image.

| Pipeline | Tool | Remark |
|---|---|---|
| Data collecting | Python | Get 400K high-resolution images |
| Preliminary data filtering | Laplacian and Sobel | Obtain subset $S$ with basic visual quality |
| Detail richness | GLCM | Obtain the set $S_G$ with rich fine-grained details |
| Content complexity | Shannon entropy | Obtain the set $S_E$ with complex and diverse content |
| Aesthetic score | LAION aesthetic predictor | Get high aesthetic score set $S_A$ |
| The final dataset | Intersection | Obtain intersection: UltraHR-100K = $S_A \cap S_E \cap S_G$ |
| UHR image caption | Gemini 2.0 | Obtain long and fine-grained descriptions for the images |

**Detail-Oriented Training Strategy for Tackling Challenge 2:** To enhance UHR detail synthesis, we propose a frequency-aware post-training method, which consists of detail-oriented timestep sampling (DOTS) and soft-weighting frequency regularization (SWFR). DOTS improves detail synthesis in UHR image generation by directing more training focus to timesteps associated with fine-grained details. Unlike the discrete and block-based decomposition approach used in Diffusion4K [16], which relies on DWT for frequency separation, our SWFR utilizes the continuous spectrum provided by the Discrete Fourier Transform (DFT) to enable more precise frequency control. By applying a soft-weighted constraint across frequency bands, SWFR encourages the model to better reconstruct high-frequency details, without compromising low-frequency structural integrity.

Through the proposed *dataset* and *training strategy*, we can *enhance* the synthesis capability of existing pre-trained T2I models [2, 3, 15, 7] in UHR image generation, with a particular focus on improving fine-grained detail representation. Furthermore, we construct a large 4K T2I benchmarks, UltraHR-eval4K (4096 × 4096), to comprehensively evaluate existing UHR generation models. Extensive experimental results demonstrate the effectiveness of our method.

## 2 Related Work

### 2.1 Text-to-Image Synthesis

Text-to-image (T2I) generation [1, 2, 3, 4, 5, 6, 7, 8, 29, 30, 31, 32, 33, 34, 35] has made notable progress owing to the emergence of diffusion-based frameworks [36, 37, 38, 39, 40, 41, 42, 43, 44, 45], which exhibit impressive ability in synthesizing visually compelling content from textual descriptions. Early methods such as Denoising Diffusion Probabilistic Models (DDPM) [46] and Denoising Diffusion Implicit Models (DDIM) [47] revealed the strength of iterative denoising procedures for producing realistic images. Subsequently, the attention to latent space diffusion [48] brought a major breakthrough, significantly lowering training complexity and enhancing scalability [2]. More recently, incorporating transformer [3, 7, 15, 49, 50] into diffusion models has further boosted image

generation quality. In this paper, we aim to enhance the generative capability of T2I models in UHR scenarios.

## 2.2 Ultra-High-Resolution Image Synthesis

UHR image generation plays a crucial role in practical domains such as industry and entertainment [16, 51, 52]. Due to computational constraints, current advanced latent diffusion models typically operate at a maximum resolution of 1024 × 1024 [2, 3, 6, 7, 8, 53, 54]. However, scaling to 4K resolution significantly increases computational demands, with cost growing quadratically with image size. Several training-free approaches have extended existing latent diffusion models for 4K generation by modifying the inference strategies of diffusion models. [13, 14, 55, 20, 21, 22, 23, 56]. DiffuseHigh [23] enhances the base-resolution generation by upscaling and subsequently re-denoising it, guided by structural information from the DWT. HiFlow [13] adopts a cascaded generation paradigm to effectively capture and utilize low-resolution flow characteristics. However, these techniques exhibit excessive smoothing, produce implausible details, and incur prolonged inference time [12]. Pixart-$\sigma$ [17] takes a pioneering step by approaching direct 4K image generation through efficient token compression in DiT. Similarly, Sana [15] introduces a cost-effective 4K generation pipeline. Despite these advancements, existing models primarily focus on training efficiency, overlooking the high-fidelity detail synthesis.

## 3 Constructing UltraHR-100K

To face the challenge of the lack of high-quality text-image pairs at UHR image generation, we construct a large-scale high-quality dataset named UltraHR-100K. We begin by collecting approximately 400K high-resolution images (with a minimum resolution of 3840×2160) using a custom Python crawler built with Scrapy, sourcing images from the web and various high-resolution imaging devices. *However, high resolution alone does not guarantee high quality. We pose a central question:* ***What constitutes a high-quality image for***

Table 2: Dataset statistical comparisons.

| Dataset | Number | Height | Width |
|---|---|---|---|
| PixArt-30k [17] | 30,000 | 1,615 | 1,801 |
| Aesthetic-4K [16] | 12,015 | 4,128 | 4,640 |
| UltraHR-100K | 104,117 | 3,648 | 5,119 |
| Aesthetic-Eval@4096 [16] | 195 | 4,912 | 6,449 |
| UltraHR-eval4K | 2,000 | 4,912 | 7,175 |

***UHR image generation?*** We argue that beyond resolution, such images should exhibit rich content complexity, fine-grained visual details, and aesthetic appeal. Accordingly, we conduct a rigorous filtering process based on three criteria—***content complexity, detail richness, and aesthetic quality***—to curate a 100K-level T2I dataset that meet these standards. The proposed UltraHR-100K provides a reliable foundation for training and evaluating models in high-fidelity UHR image generation. The data processing pipeline is provided in Table 1.

**Preliminary Data Filtering.** High-resolution images scraped from the web often suffer from blur, noise, or lack of texture, which can significantly degrade image quality. To eliminate such artifacts, we apply a two-stage low-level quality filter. First, we compute the Laplacian variance to assess image sharpness and discard samples below a blur threshold. Second, we apply the Sobel operator to measure edge density, removing overly flat or textureless images. This process yields a cleaned subset $S$ with sufficient basic visual quality.

**Detail Richness.** Fine-grained details are essential for training generative models to preserve high-frequency content. To quantify the aspect, we compute Gray-Level Co-occurrence Matrix (GLCM) score, including contrast, entropy, and correlation across multiple directions. These metrics capture spatial pixel relationships indicative of texture complexity. We then select the top 50% of images from $S$ with the highest aggregated GLCM scores, resulting in subset $S_G$.

**Content Complexity.** Visually complex images and diverse spatial structures are more valuable for guiding generation models to achieve rich content. We use Shannon entropy as a proxy to measure the content complexity of each image. Images with higher entropy tend to contain more varied pixel intensities. From subset $S$, we retain the top 50% highest-entropy images to construct subset $S_E$.

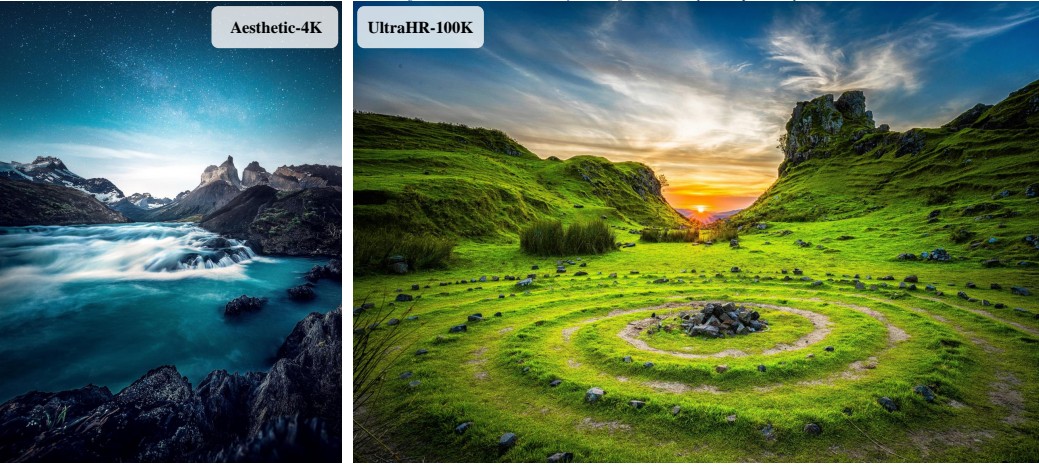

*Turquoise waters cascade over rocky outcrops, surrounded by rugged mountains capped with snow under a starry sky.*

*A vibrant landscape scene unfolds with a stone circle prominently displayed in a verdant valley, framed by lush green hills and a striking sunset illuminating the horizon. The stone circle, with rocks neatly arranged in concentric patterns around a central pile, draws the eye amidst the open, grassy field. The surrounding hills rise gently, their slopes covered in a thick carpet of bright green vegetation, while rugged rock foarmations punctuate the landscape. The warm hues of the setting sun cast a golden glow, contrasting with the cool blues in the sky, creating a serene and picturesque atmosphere.*

**Aesthetic-4K**  **UltraHR-100K**

Figure 4: Comparison between our UltraHR-100K and Aesthetic-4K[16]. Captions in our UltraHR-100K provide more expressive descriptions, encompassing not only global summaries of the image content but also rich details that enhance semantic alignment.

**Aesthetic Quality.** Aesthetic appeal is an important factor in image realism and human preference. To incorporate this dimension, we adopt the LAION Aesthetic Predictor [57], a neural network trained to estimate perceptual quality. It outputs a scalar score reflecting visual composition, color harmony, and overall appeal. We rank all images in $S$ by their aesthetic scores and retain the top 50% to form subset $S_A$, consisting of the most visually pleasing samples.

**UltraHR-100K.** To ensure that the final dataset consists of high-quality UHR images with *diverse content, rich textures, and strong aesthetic appeal*, we take the intersection of the three selected subsets. Specifically, the final dataset is defined as:

$$\text{UltraHR-100K} = S_G \cap S_E \cap S_A \tag{1}$$

This intersection guarantees that each image in UltraHR-100K simultaneously meets high standards in detail richness, content complexity, and aesthetic quality, as shown in Figure 2. In addition, we construct a evaluation subset from our dataset—**UltraHR-eval4K**—containing 2,000 images. Table 2 compares our UltraHR-100K with Aesthetic-4K [16] and PixArt-30K [17]. These statistics highlight that UltraHR-100K not only improves dataset scale, but also provides more extensive spatial content.

**UHR Image Caption.** UHR images typically contain significantly more visual information than standard-resolution images, making them inherently more semantically complex. However, existing datasets [16, 57] often provide only short captions, limiting the semantic expressiveness of generative models. To address this issue, we leverage Gemini 2.0 [28], a state-of-the-art commercial vision-language model (VLM), to generate rich and detailed captions for our dataset. As illustrated in Figures 3 and 4, our captions are not only substantially longer but also encompass both global summaries and fine-grained descriptions, enhancing alignment with complex image content.

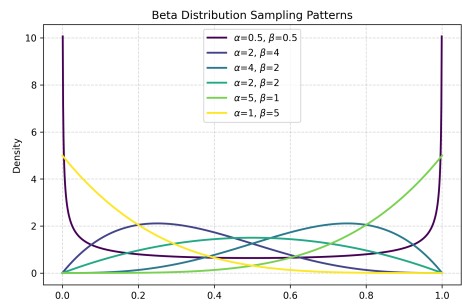

Figure 5: Relation between weighting ratio and timesteps with beta sampling strategy.

# 4 Frequency-Aware Post-Training

Pretrained T2I models, trained on large-scale datasets, exhibit strong capabilities in semantic and content planning. However, they often struggle to synthesize fine-grained details when extended to UHR scenarios [12, 15]. In this work, we focus on enhancing the detail synthesis ability of pretrained T2I models through tailored post-training strategies. To this end, we propose a frequency-aware post-training method (FAPT). Specifically, FAPT consists of two parts: detail-oriented timestep sampling (DOTS) and soft-weighting frequency regularization (SWFR). DOTS improves detail synthesis in UHR image generation by directing more training focus to timesteps associated with fine-grained details. Meanwhile, SWFR imposes a soft-weighted constraint across the frequency spectrum, guiding the model to better preserve and reconstruct high-frequency details.

## 4.1 Detail-Oriented Timestep Sampling

**Motivation.** Existing study [58] have validated the observation that the overall image structure (low-frequency signals) is largely reconstructed in the early denoising steps, while fine-grained details (high-frequency signals) are progressively synthesized in the later stages of the denoising process. This insight motivates us to design a sampling strategy that emphasizes the later stages of the denoising process, aiming to enhance the learning of fine-grained details during post-training stage.

**DOTS.** To achieve this target, we adopt a beta sampling strategy, which provides a simple yet flexible mechanism to bias the sampling distribution over denoising timesteps, as shown in Figure 5. Specifically, we first draw a timestep $t \in (0, 1)$ from a Beta distribution parameterized by shape parameters $\alpha$ and $\beta$:

$$t \sim \text{Beta}(\alpha, \beta). \tag{2}$$

The Beta distribution yields a rich family of unimodal or skewed distributions over the interval $(0, 1)$, and its probability density function is given by:

$$\pi_{\text{beta}}(t; \alpha, \beta) = \frac{1}{\text{B}(\alpha, \beta)} t^{\alpha-1} (1-t)^{\beta-1}, \tag{3}$$

where $\text{B}(\alpha, \beta) = \frac{\Gamma(\alpha)\Gamma(\beta)}{\Gamma(\alpha+\beta)}$ is the Beta function. By adjusting $\alpha$ and $\beta$, we can control the bias of the sampling distribution. This sampling mechanism naturally supports adaptive emphasis in training: by emphasizing later denoising timesteps, we can guide the model to focus on high-frequency details.

## 4.2 Soft-Weighting Frequency Regularization

**Motivation.** Large pre-trained T2I models [2, 3, 5, 7] demonstrate strong semantic planning from diverse data exposure but struggle with fine-grained detail synthesis in UHR scenarios. Existing UHR T2I models focus mainly on training efficiency [15, 17], often neglecting high-fidelity detail. Diffusion4K [16] introduces DWT-based frequency decomposition to enable 4K training, yet DWT yields coarse and discontinuous frequency separation, limiting its effectiveness for UHR modeling. To overcome this, we adopt DFT-based decomposition, which provides finer, globally coherent frequency representations better suited for capturing fine-scale structures in high-resolution synthesis.

**SWFR.** To enhance fine-scale fidelity in UHR image synthesis, we introduce a soft-weighting frequency regularization that complements the standard diffusion loss by explicitly supervising frequency consistency, with an emphasis on high-frequency components. Formally, consider the standard diffusion process:

$$\boldsymbol{z}_t = \alpha_t \cdot \boldsymbol{x}_0 + \sigma_t \cdot \boldsymbol{\epsilon}, \tag{4}$$

where $\boldsymbol{x}_0$ denotes the data distribution, $\boldsymbol{\epsilon}$ is sampled from standard normal distribution, and $\alpha_t$, $\sigma_t$ are known coefficients in the diffusion formulation. Recent T2I models [5, 7, 15] adopt rectified flows to predict velocity $\boldsymbol{v}$, with the objective as follows:

$$\boldsymbol{v}_\Theta(\boldsymbol{z}_t, t) = \boldsymbol{\epsilon} - \boldsymbol{x}_0. \tag{5}$$

To regularize the model in the frequency domain, we compute the 2D Discrete Fourier Transforms (DFT) of both prediction $\boldsymbol{x}$ and target $\boldsymbol{y}$:

$$\hat{\boldsymbol{x}} = \mathcal{F}(\boldsymbol{x}), \quad \hat{\boldsymbol{y}} = \mathcal{F}(\boldsymbol{y}), \tag{6}$$

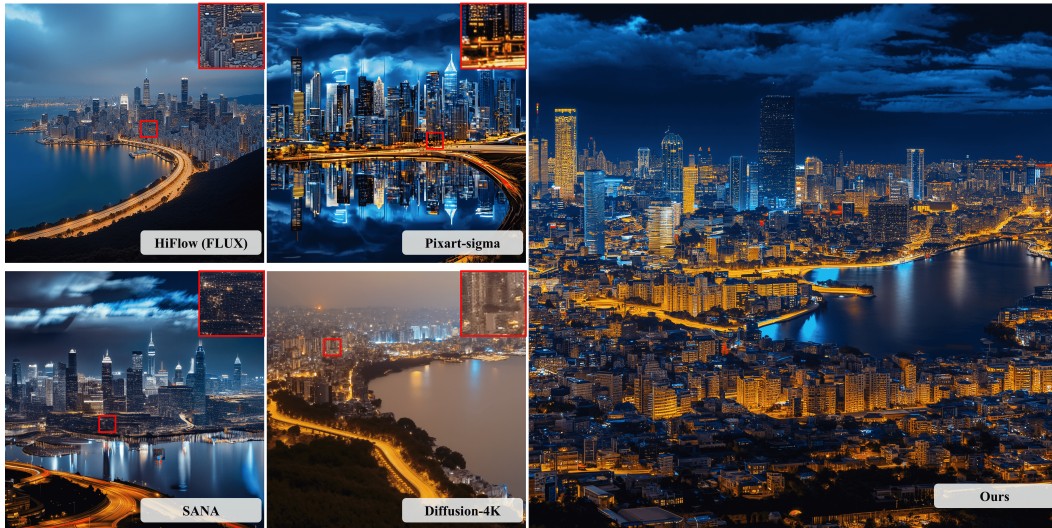

Figure 6: Qualitative comparisons with SOTA methods on our UltraHR-eval4K ($4096\times 4096$). Compared with previous works, our method is capable of generating visually complex images with rich semantic content. More visual examples are available in the supplementary materials.

where $\mathcal{F}(\cdot)$ denotes the DFT. Let $x$ and $y$ denote the model prediction (e.g., $x = v_\Theta(z_t, t)$) and target (e.g., $y = \epsilon - x_0$), respectively. We define a frequency-domain regularization term as:

$$\mathcal{L}_{\text{freq}} = \mathbb{E}\left[|w(r) \cdot \hat{x} - w(r) \cdot \hat{y}|^2\right], \tag{7}$$

where $w(r)$ is a frequency soft weighting function designed to boost high-frequency supervision:

$$w(r) = 1 + \lambda \cdot \frac{\exp(\gamma r) - 1}{\exp(\gamma) - 1}, \quad r \in [0, 1], \tag{8}$$

and $r$ is the normalized distance from the center of the frequency plane. Hyperparameters $\lambda$ and $\gamma$ control the strength and steepness of high-frequency emphasis, respectively. Finally, the overall training objective is defined as:

$$\mathcal{L}_{\text{total}} = \mathcal{L}_{\text{diff}} + \lambda_{\text{freq}} \cdot \mathcal{L}_{\text{freq}}, \tag{9}$$

where $\mathcal{L}_{\text{diff}}$ denotes the diffusion loss, which can be instantiated as velocity prediction loss ($\|v_\Theta(z_t, t) - (\epsilon - x_0)\|^2$). $\lambda_{\text{freq}}$ is a balancing coefficient that controls the strength of frequency-domain supervision. This regularization $\mathcal{L}_{\text{freq}}$ encourages the model to maintain consistent spectral power between prediction and target, especially in high-frequency bands.

## 5 Experiments

### 5.1 Implementation Details

**Overall Training Setting.** We adopt a two-stage training strategy. In the first stage, we follow the Logit-Normal Sampling scheme introduced in SD3 [6] and perform fine-tuning on our UltraHR100K dataset, aiming to enhance the semantic planning capability in UHR generation. In the second stage, we apply our proposed frequency-aware post-training method, which focuses on high-frequency learning to further improve the fine-grained details. We use the CAMEWrapper [15] optimizer with a constant learning rate of 1e-4, and employ mixed-precision training with a batch size of 24. The first-stage training is conducted for 4K iterations, followed by 8K iterations in the second stage. Due to computational constraints, we conduct training solely on SANA, and all experiments are performed on four H20 GPUs.

**Baselines.** To comprehensively evaluate our approach, we conduct extensive comparisons against SOTA methods for UHR image generation, which can be broadly categorized into three groups. The first group consists of powerful T2I models combined with super-resolution technique, BSRGAN [59]. The second group includes training-free approaches, where we evaluate FLUX[7]) using corresponding training-free generation methods, I-Max [60] and HiFlow [13]. Lastly, we compare with leading

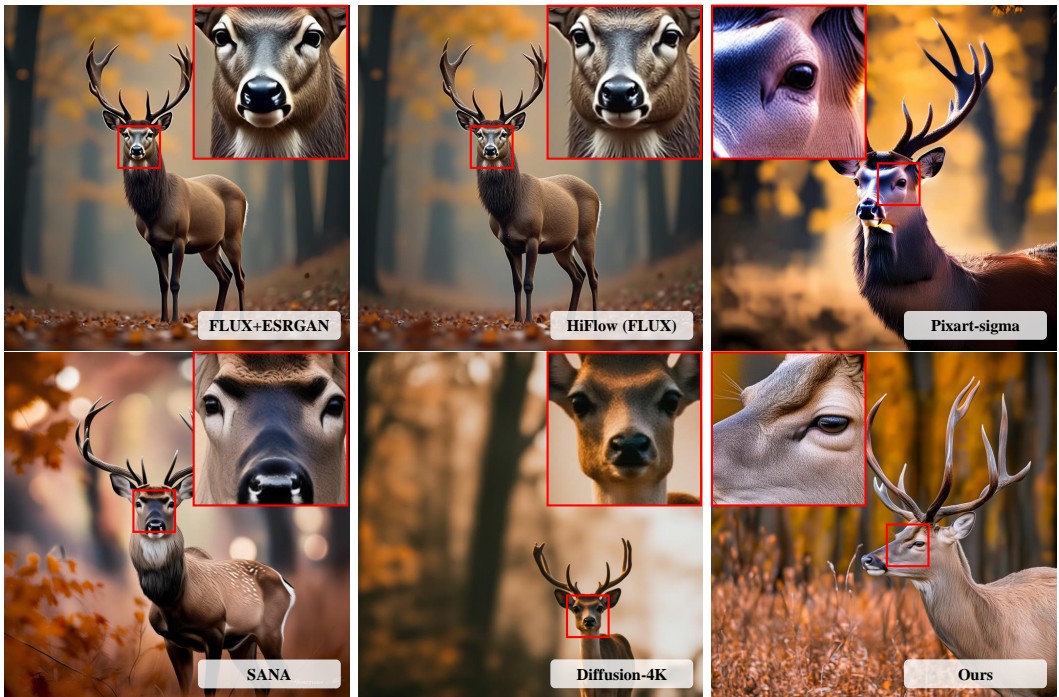

Figure 7: Qualitative comparisons with SOTA methods on our UltraHR-eval4K (4096× 4096). Compared with previous works, our method can generate realistic textures and fine-grained details.

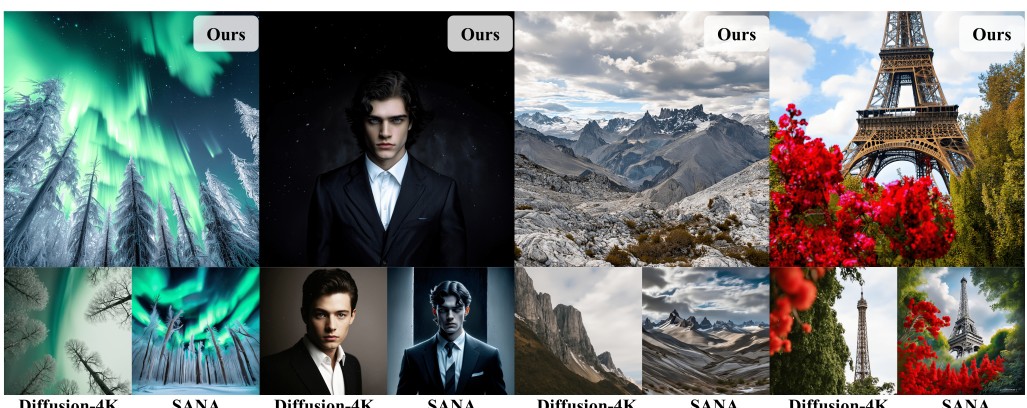

Figure 8: More visual comparisons demonstrate that our method consistently produces high-quality results. Additional and more diverse comparisons can be found in the supplementary material.

training-based UHR generation models, including Pixart-$\sigma$ [17], SANA [15], and Diffusion4K [16]. All baselines are evaluated under their official settings to ensure a fair and consistent comparison.

**Evaluation.** We employ several metrics to assess the quality of the generated images, with a particular focus on our evaluation sets, UltraHR-eval4K. To evaluate image-text consistency, we calculate the long CLIP score [61] and Fine-Grained (FG) CLIP score [62]. Additionally, the Fréchet Inception Distance (FID) [63] and Inception Score (IS) [64] are computed to evaluate the overall image quality of the generated images. Following previous works [13, 12], we compute the FID-patch and IS-patch to evaluate the local quality and details of the images, which are based on local image patches. These metrics provide a comprehensive evaluation of the overall quality, detail retention in the generated images.

Table 3: **Quantitative comparison with other baselines on our UltraHR-eval4K ($4096 \times 4096$) benchmark.** The best result is highlighted in **bold**.

| Method | FID ↓ | FID$_{patch}$ ↓ | IS ↑ | IS$_{patch}$ ↑ | CLIP ↑ | FG-CLIP ↑ |
|---|---|---|---|---|---|---|
| FLUX [7] + BSRGAN [59] | 37.651 | 43.143 | 11.773 | 5.389 | 31.45 | 28.02 |
| SD3.5 [6] + BSRGAN [59] | 31.870 | 25.598 | 12.780 | 5.456 | 31.75 | 28.66 |
| I-Max(FLUX) [60] | 37.667 | 37.835 | 11.991 | 4.391 | 31.49 | 27.78 |
| HiFlow(FLUX) [13] | 35.892 | 38.327 | 11.767 | 4.620 | 31.52 | 27.75 |
| Pixart-$\sigma$ [17] | 33.171 | 32.198 | 12.212 | 5.390 | 31.78 | 28.65 |
| SANA [15] | 37.070 | 38.795 | 11.778 | **5.649** | 31.70 | 28.60 |
| Diffusion4K [16] | 39.857 | 38.515 | 10.832 | 3.235 | 31.41 | 26.48 |
| **Ours(UltraHR-100K)** | 33.995 | 20.932 | 12.502 | 5.020 | **31.85** | 28.65 |
| **Ours(UltraHR-100K+FAPT)** | **31.748** | **15.795** | **12.995** | 5.104 | 31.82 | **28.68** |

Table 4: Left: User study results conducted on our UltraHR-eval4K. Right: Quantitative comparison on Aesthetic-Eval@4096. The results demonstrate the superior performance of our method.

| Method | Overall Quality | Detail Quality | Text-Image Alignment | Preference |
|---|---|---|---|---|
| Pixart-$\sigma$ [17] | 14% | 10% | 16% | 18% |
| SANA [15] | 4% | 8% | 8% | 6% |
| Diffusion4K [16] | 12% | 4% | 6% | 6% |
| **Ours** | **70%** | **78%** | **72%** | **70%** |

| Method | FID ↓ | FID$_{patch}$ ↓ | CLIP ↑ | FG-CLIP ↑ |
|---|---|---|---|---|
| Pixart-$\sigma$ [17] | 150.593 | 44.702 | 34.88 | 28.48 |
| SANA [15] | 146.027 | 37.031 | 34.62 | 28.61 |
| Diffusion4K [16] | 152.790 | 39.729 | 33.99 | 26.06 |
| **Ours** | **142.965** | **24.008** | **35.08** | **28.64** |

## 5.2 Comparison to State-of-the-Art Methods

**Quantitative Comparison.** Table 3 summarizes the quantitative performance on our UltraHR-eval4K benchmark ($4096 \times 4096$). Our method consistently achieves superior scores on key perceptual metrics such as FID, FID-patch and IS, indicating its strong capability in generating high-quality images with fine-grained textures. Moreover, our method achieves competitive CLIP scores, reflecting its ability to maintain semantic alignment with the input prompt. Notably, our method yields a substantial improvement in FID$_{patch}$, highlighting its effectiveness in synthesizing fine-grained details. This result demonstrates that our proposed approach significantly enhances the detail generation capability of pre-trained T2I models in UHR scenarios.

**Qualitative Comparison.** Figure 6 presents qualitative comparisons on UltraHR-eval4K ($4096 \times 4096$), focusing on the overall semantic richness and spatial layout of the generated images. While existing SOTA methods struggle to produce coherent and content-rich scenes at such ultra-high resolution, our method demonstrates a strong capability in generating visually complex images with diverse and semantically meaningful elements. This highlights our model's superior capacity for global spatial reasoning and semantic planning in large-scale synthesis. In Figure 7, we further compare fine-grained textures and local details. Our method produces sharper structures and more realistic textures, faithfully preserving high-frequency information that other methods tend to miss or oversmooth. These results collectively demonstrate the effectiveness of our proposes dataset and method in enhancing both the global semantics and local fidelity for ultra-high-resolution text-to-image generation. Figure 8 presents more visual comparisons.

**User Study.** As shown in Table 4, we conducted a user study with 5 volunteers evaluating 50 randomly selected cases. Images were rated on overall quality, detail quality, text-image alignment and preference. The results demonstrate the superiority of our method across all aspects.

**Comparisons on Public Benchmark.** We conduct a quantitative comparison on the publicly available Aesthetic-4K benchmark, specifically the Aesthetic-Eval@4096 subset, as reported in Table 4. This evaluation set contains 195 image-text pairs, where all images have a short side greater than 4096 pixels. Due to the limited number of samples, the reported FID scores are relatively high. Nonetheless, the results clearly demonstrate the superior performance of our method, supporting its robustness and generalizability beyond our proposed benchmark.

Table 5: Ablation study of our key components and data scale. Model A is a baseline using full fine-tuning on our dataset. The comparison between C (trained on a partial dataset) and D (full dataset) validates the effectiveness of large-scale data.

| Model | DOTS | SWFR | Dataset | FID ↓ | FID$_{patch}$ ↓ | CLIP ↑ |
|-------|------|------|---------|-------|-----------------|--------|
| LoRA | × | × | Full | 35.07 | 35.02 | 31.80 |
| A | × | × | Full | 33.99 | 20.93 | **31.85** |
| B | ✓ | × | Full | 32.57 | 19.95 | 31.79 |
| C | ✓ | ✓ | Part | 32.75 | 18.42 | 31.81 |
| D | ✓ | ✓ | Full | **31.74** | **15.79** | 31.82 |

## 5.3 Ablation Study

We conduct a comprehensive ablation study to validate the effectiveness of our proposed training strategy and the importance of large-scale data. As shown in Table 5, Model A serves as the baseline without our proposed DOTS and SWFR. Model B introduces DOTS, resulting in consistent improvements in both FID and patch-level FID, demonstrating its effectiveness in guiding the sampling process. Further incorporating SWFR (Model D) yields substantial improvements, particularly in patch-level FID, confirming that our proposed regularization enhances the detail synthesis capability of T2I models. To evaluate the impact of training data scale, we compare Model C and Model D. Model C is trained with a randomly sampled 15K subset of our UltraHR-100K using the same training strategy. The performance drop compared to Model D clearly highlights the importance of large-scale UHR data in achieving high-fidelity and semantically aligned image generation.

**Analysis for DOTS.** The DOTS module employs a Beta($\alpha$, $\beta$) distribution to guide timestep sampling, where $\alpha$ and $\beta$ control the bias along the denoising trajectory. When $\alpha < \beta$, sampling favors later steps (near $t = 0$) that refine high-frequency details; when $\alpha > \beta$, it leans toward early steps (near $t = 1$) emphasizing global structure. In our experiments, we set $\alpha = 2$, $\beta = 4$, biasing sampling toward later steps to better capture fine details crucial for ultra-high-resolution generation. An ablation study (Table

Table 6: Analysis for DOTS.

| Method | FID | FID_patch | CLIP |
|--------|-----|-----------|------|
| ($\alpha = 3, \beta = 4$) | 33.196 | 22.143 | 31.83 |
| ($\alpha = 1, \beta = 4$) | 33.727 | 25.095 | 31.79 |
| ($\alpha = 2, \beta = 5$) | 33.874 | 23.850 | 31.82 |
| ($\alpha = 2, \beta = 3$) | 33.638 | 24.638 | **31.84** |
| ($\alpha = 2, \beta = 4$) | **31.748** | **15.795** | 31.82 |

6) varying $\alpha$ and $\beta$ confirms this choice: larger $\alpha$ weakens detail learning, smaller $\alpha$ harms semantic consistency, and overly concentrated or flattened distributions reduce diversity. These results validate ($\alpha = 2, \beta = 4$) as a balanced and effective configuration.

## 6 Conclusion

In this paper, we present UltraHR-100K, a curated dataset of 100K UHR images with rich textual annotations. Each image is carefully selected to ensure high levels of detail, visual complexity, and aesthetic appeal. Moreover, we introduce a frequency-aware post-training method, which includes: (i) Detail-Oriented Timestep Sampling (DOTS), and (ii) Soft-Weighting Frequency Regularization (SWFR). Experiments on our proposed UltraHR-eval4K benchmark confirm that our approach significantly boosts both the visual fidelity and fine-detail accuracy of UHR image synthesis.

**Limitations and future works.** Our main limitations lie in two aspects. First, while the proposed frequency-aware post-training strategy effectively enhances fine-detail synthesis, it introduces a slight degradation in text–image alignment, as shown in Table 5. Second, our dataset currently contains a relatively limited amount of portrait data, which constrains the improvement in ultra-high-resolution (UHR) portrait generation, as illustrated in Figure 8. In future work, we plan to develop more balanced training strategies to alleviate the alignment issue and expand our dataset with additional high-quality UHR portrait images to further improve performance in portrait synthesis.

**Acknowledgments.** This work was supported by Natural Science Foundation of China: No. 62406135, Natural Science Foundation of Jiangsu Province: BK20241198, and Gusu Innovation and Entrepreneur Leading Talents: No. ZXL2024362.

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
