# OpenReview forum: "UltraHR-100K: Enhancing UHR Image Synthesis with A Large-Scale High-Quality Dataset"
_NeurIPS.cc/2025/Conference — NeurIPS 2025 poster_

### Official Review · Reviewer_UXyJ · 2025-06-16

**Clarity:** 3
**Significance:** 2
**Originality:** 3
**Rating:** 4
**Confidence:** 2

**Summary:**

This paper addresses the challenges in ultra-high-resolution (UHR) text-to-image generation, namely the lack of large-scale high-quality datasets and tailored training strategies for fine-grained detail synthesis. The authors construct a comprehensive dataset, emphasizing its rigorous curation based on detail richness, content complexity, and aesthetic quality. They also propose a frequency-aware post-training method, comprising Detail-Oriented Timestep Sampling (DOTS) and Soft-Weighting Frequency Regularization (SWFR), to enhance high-frequency detail preservation in diffusion models.

**Questions:**

Please see the weaknesses part.

**Ethical Concerns:**

["NO or VERY MINOR ethics concerns only"]

**Final Justification:**

I appreciate the authors' detailed response, but I maintain my original assessment for the following reasons:

1. While the authors emphasize their frequency-aware post-training strategy, the core innovation remains dataset-centric. The proposed DOTS and SWFR modules, though useful, are incremental improvements rather than paradigm-shifting contributions. As noted in my initial review, this work would be better suited for the Datasets & Benchmarks Track.
2. Regarding Q2, the authors do not directly address my question, and they talk about some other issues.

So I keep my initial rating.

**Limitations:**

yes

**Quality:**

3

**Strengths And Weaknesses:**

[Strengths]
1. This paper constructs a large-scale text-to-image dataset with high-quality images and texts.
2. The model trained based on the proposed dataset demonstrates favorable performance.


[Weaknesses]
1. The vast majority of this paper's contributions lie in the proposal of a large dataset, with limited technical innovations and more engineering-oriented content. It seems more appropriate for submission to the NeurIPS Datasets & Benchmarks Track rather than the main conference.
2. As a dataset-focused paper, the comparison between the proposed dataset and existing ones is rather superficial. Additionally, it is recommended to add citations for the comparative methods listed in Table 2.
3. Regarding the DOTS module proposed in Section 4, the paper lacks clarity on the specific strategies for selecting α and β, as well as the parameter sensitivity analysis.
4. Is Google Imagen a comparable approach to the proposed method? If so, it is not discussed in the paper.
5. It is suggested to use vector graphics for figures, such as Figure 1.

---

> ### Author Rebuttal · Authors · 2025-07-28
>
> ## Q1: The vast majority of this paper's contributions lie in the proposal of a large dataset, with limited technical innovations and more engineering-oriented content.
>
> Answer: We respectfully disagree with the reviewer’s assessment that our work is primarily a dataset paper with limited technical contributions. While the construction of the UltraHR dataset is indeed a key component of our work, our paper also proposes a **novel and effective frequency-aware post-training strategy** tailored for ultra-high-resolution (UHR) text-to-image generation. This algorithmic contribution is clearly distinct from the dataset itself and has been positively acknowledged by other reviewers.
>
> For instance, **Reviewer 4Jyg** noted:
> > *“Helping models to capture the details in generation also makes sense. A new post-training strategy is proposed to compile the new dataset.”*
>
> Similarly, **Reviewer WG7N** highlighted:
> > *“The proposed post-training method is well-motivated and technically sound, with strong rationale provided for the introduced components.”*
>
> and further emphasized that
>
> > *“The post-training method introduced offers a relatively new approach to improving text-to-image generation in the ultra-high-resolution regime.”*
>
> **Reviewer 1nz3** also affirmed the value of our method:
> > *“Leveraging the proposed frequency-aware post-training strategy, the model achieves strong performance in UHR image synthesis.”*
>
> Although our method is relatively simple, we believe that **simplicity does not undermine novelty or impact**. On the contrary, **simple yet effective approaches often demonstrate greater potential for scalability, reproducibility, and broader adoption** within the community.
>
> We therefore believe our work presents both a **significant dataset** and a **technically novel contribution**, which together offer substantial value to the **main NeurIPS conference**.
>
> ---
>
> ## Q2: As a dataset-focused paper, the comparison between the proposed dataset and existing ones is rather superficial.
>
> Answer:  We thank the reviewer for the valuable feedback and apologize for the missing citations in Table 2. We will add all relevant references in the revised version to ensure proper attribution and clarity.
>
> Regarding the comparison with existing datasets, we respectfully clarify that our UltraHR-100K dataset presents **significant improvements** over previous datasets in terms of ** resolution, scale, and diversity** . Beyond basic comparison, we also introduce a **novel, task-specific pipeline for curating high-quality ultra-high-resolution (UHR) data**, which is a key contribution of our work. This pipeline incorporates multi-dimensional filtering and tailored captioning strategies that are specifically designed to address the unique challenges of UHR image-text data curation.
>
> The motivation and significance of our dataset have been well recognized by other reviewers. For example, **Reviewer 4Jyg** noted:
> > *"The motivation of proposing UHR100K is straight-forward and intuitive. Instead of using complex training-free methods, which still struggle generating detail-rich images, the authors curate more and better data as a simple solution."*
>
> **Reviewer WG7N** emphasized:
> > *"The methodology for constructing the dataset is carefully considered and well-documented. The filtering and captioning strategies demonstrate a rigorous approach to dataset curation."*
> > *"The proposed UHR dataset has the potential to serve as a valuable resource for the community. High-quality, caption-rich ultra-high-resolution image datasets remain scarce, and this contribution could help advance research in text-to-image generation at scale."*
>
> **Reviewer 1nz3** also acknowledged the strength of our dataset:
> > *"The paper presents a rigorously curated UHR dataset (UltraHR-100K) with significantly larger scale and higher quality than previous datasets such as Aesthetic-4K. The multi-dimensional filtering process is clearly motivated and ensures both diversity and fidelity of the data."*
>
> Given the above, we believe that **UltraHR-100K will make a substantial contribution to the field of ultra-high-resolution image synthesis and serve as a valuable open resource for the community.**
>
> ---
>
> ## Q3: Regarding the DOTS module proposed in Section 4, the paper lacks clarity on the specific strategies for selecting α and β, as well as the parameter sensitivity analysis.
>
> Answer: We thank the reviewer for pointing this out. We apologize for the lack of detailed explanation in the main paper and have included a more comprehensive description in the supplementary materials. Specifically, the DOTS module utilizes a **Beta(α, β)** distribution to guide the timestep sampling process. The parameters α and β are used to control the sampling bias toward different phases of the denoising trajectory:
>
> - When **α < β**, the distribution favors **later denoising steps** (closer to *t = 0*), which tend to focus on **high-frequency detail refinement**.
> - When **α > β**, the sampling shifts toward **early denoising steps** (closer to *t = 1*), which contributes more to **global semantic structure**.
>
> In our main experiments, we set **α = 2** and **β = 4**, intentionally biasing the sampling toward later timesteps to better capture fine details, which are critical for ultra-high-resolution image generation.
>
> To further investigate the sensitivity of α and β, we conducted an ablation study, as shown in **Table 1**. Below is a summary of our findings:
>
> - **(α = 3, β =4)**: Increasing α shifts the distribution rightward, leading to more early-step sampling. This **negatively impacts high-frequency learning**, as the model receives less supervision in later steps.
> - **(α =1, β =4)**: Decreasing α moves the distribution leftward, favoring later-step sampling. This can **reduce semantic fidelity**, as the model may not learn sufficient global structure.
> - **(α =2, β =5)**: Increases the distribution’s peak near the center, causing **sampling to concentrate too narrowly**, limiting diversity in supervision.
> - **(α =2, β =3)**: Lowers the distribution peak, increasing sampling across both early and late steps. While more balanced, this can **dilute the benefits** of our targeted sampling strategy.
>
> These results support our choice of **(α = 2, β = 4)** as a **balanced and effective configuration** that encourages detail learning without sacrificing semantic structure. We will include these discussions and the corresponding table in the revised paper to improve clarity.
>
> ---
>
> ## Q4: Is Google Imagen a comparable approach to the proposed method? If so, it is not discussed in the paper.
>
> Answer: We appreciate the reviewer’s suggestion regarding Google Imagen. While Google Imagen is indeed a powerful closed-source text-to-image generation model, it is not directly comparable to our proposed method for several reasons:
> 1）Imagen, as described in the original paper **[1]**, focuses on generating 256×256 or 1024×1024 images. However, it does not natively support ultra-high-resolution generation (e.g., ≥ 4K resolution), which is the primary focus of our work.
> 2）Imagen is not publicly released, making it infeasible to evaluate on our UltraHR benchmarks or to integrate into our framework for a fair comparison. As a result, a quantitative or qualitative comparison is currently inaccessible to the research community. We will clarify this distinction in the updated version and will consider mentioning Imagen explicitly to avoid confusion.
>
> [1] Saharia, Chitwan, et al. "Photorealistic text-to-image diffusion models with deep language understanding." Advances in neural information processing systems 35 (2022): 36479-36494.
>
> ---
>
> ## Q5: It is suggested to use vector graphics for figures, such as Figure 1.
>
> Answer: We thank the reviewer for pointing this out and sincerely apologize for the current rasterized version of Figure 1. We fully agree that vector graphics would improve visual clarity and readability. We will update all relevant figures, including Figure 1, to vector format (e.g., PDF or SVG) in the revised version to ensure better scalability and presentation quality.
>
> ---
>
> Table 1. Ablation study for DOTS.
> | Method  | FID   | FID_patch      | CLIP   |
> |---------|--------|--------|--------|
> |**(α = 3, β =4)** |  33.196  | 22.143  | 31.83 |
> |**(α = 1, β =4)** | 33.727 | 25.095  | 31.79 |
> |**(α = 2, β =5)** |   33.874 |  23.850 | 31.82 |
> |**(α = 2, β =3)** | 33.638 | 24.638 |  **31.84** |
> | Ours-**(α = 2, β =4)**   | **31.748** | **15.795** |31.82  |

---

> > ### Author Response · Authors · 2025-08-07
> > **Respond to Reviewer UXyJ**
> >
> > Thank you very much for your valuable comments, which have greatly contributed to enhancing the quality of our paper. We would like to inquire if there are any remaining questions or concerns regarding our work. If so, we will do our utmost to address and clarify them.

---

> > > ### Comment · Reviewer_UXyJ · 2025-08-07
> > >
> > > Thank you for your reply. I have little confidence in this field, as it feels like work centered around datasets. However, the overall work is solid.

---

> > > > ### Author Response · Authors · 2025-08-08
> > > > **Respond to Reviewer UXyJ**
> > > >
> > > > Thank you very much for your thoughtful feedback and for recognizing the solidity of our work. We truly appreciate your comments. We will carefully integrate the suggestions and improvements from the rebuttal into the final version of the paper. If you have any further questions or suggestions, please do not hesitate to let us know. We are more than willing to address any remaining issues and ensure the clarity and quality of our work. Thank you again for your time and valuable comments.

---

> ### Author Response · Authors · 2025-08-05
> **Respond to Reviewer UXyJ**
>
> Dear Reviewer Reviewer UXyJ,
>
> As the discussion deadline approaches [Aug 06 '25 (AoE)], we fully understand that you may have other pressing commitments. We are genuinely grateful for the time and effort you have dedicated to reviewing our submission, as well as for your insightful and constructive feedback. The discussion process not only provides an invaluable opportunity to improve the quality of the paper for the authors, but also deepens the reviewers' understanding of the work, contributing to the overall advancement of the academic community. We sincerely hope to continue addressing any unresolved concerns through our responses. Thank you once again for your invaluable time and for your role in helping us refine and strengthen our work!

---

### Official Review · Reviewer_1nz3 · 2025-06-23

**Clarity:** 4
**Significance:** 4
**Originality:** 4
**Rating:** 5
**Confidence:** 4

**Summary:**

This paper introduces UltraHR-100K, a large-scale dataset designed for ultra-high-resolution (UHR) text-to-image synthesis, consisting of 100K images with resolutions exceeding 3K. To address the challenge of fine-grained detail synthesis in UHR scenarios, the authors propose a frequency-aware post-training strategy that includes Detail-Oriented Timestep Sampling (DOTS) and Soft-Weighting Frequency Regularization (SWFR). Experimental results demonstrate the effectiveness of the proposed method, although some missing comparisons and qualitative studies could further strengthen the work.

**Questions:**

Please refer to the Weaknesses section for potential directions to enhance the paper. While these additions are not essential for acceptance, addressing them could improve the paper’s completeness and persuasiveness.

**Ethical Concerns:**

["NO or VERY MINOR ethics concerns only"]

**Final Justification:**

I have read the rebuttal and appreciate the authors’ clarifications. They have addressed my main concerns, and I will keep my score as 5.

**Limitations:**

yes

**Quality:**

4

**Strengths And Weaknesses:**

Strengths
1. The paper presents a rigorously curated UHR dataset (UltraHR-100K) with significantly larger scale and higher quality than previous datasets such as Aesthetic-4K. The multi-dimensional filtering process (GLCM-based texture analysis, Shannon entropy, and aesthetic score) is clearly motivated and ensures both diversity and fidelity of the data.
2. Leveraging the proposed frequency-aware post-training strategy, the model achieves strong performance in UHR image synthesis. The qualitative results (e.g., Figure 6) are visually compelling and show clear improvements over baselines.

Weaknesses
1. The evaluation is conducted exclusively on the authors’ own UltraHR-eval4K benchmark. Including comparisons on established public datasets such as Aesthetic-4K would provide stronger and more objective evidence of the method’s generalizability and robustness, since hyperparameter tuning can significantly influence performance.
2. The ablation studies are entirely quantitative. Incorporating qualitative ablation would offer more intuitive insights into how each component contributes to improved visual quality.

---

> ### Author Rebuttal · Authors · 2025-07-28
>
> Firstly, it is a great honor to receive such high praise. We appreciate this helpful suggestion.
>
> ## For the comparisons on established public datasets
> We conduct a quantitative comparison on the publicly available Aesthetic-4K benchmark, specifically the Aesthetic-Eval@4096 subset, as reported in Table 1. This evaluation set contains 195 image-text pairs, where all images have a short side greater than 4096 pixels. Due to the limited number of samples, the reported FID scores are relatively high. Nonetheless, the results clearly demonstrate the superior performance of our method, supporting its robustness and generalizability beyond our proposed benchmark.
>
> ---
>
> ## For the qualitative ablation
> We apologize for not including qualitative results in the ablation study. Due to NeurIPS rebuttal guidelines, we are unable to provide visualizations in the response. However, we will include intuitive qualitative comparisons of different ablation variants in the revised version of the paper to better illustrate the contribution of each component.
>
> ---
>
> Table 1. Quantitative comparison on Aesthetic-Eval@4096.
> We report FID, FID_patch, CLIP, and FG-CLIP metrics. The results demonstrate the superior performance of our model on this public UHD benchmark.
>
> | Method  |    FID     | FID_patch    | CLIP  | FG-CLIP |
> |-------------|-------------|-------------|---------------|-------------------|
> | Pixart-σ    | 150.593 | 44.702  | 34.88 | 28.48  |
> | SANA    | 146.027 | 37.031  | 34.62 | 28.61  |
> | DIFF4K  | 152.790 | 39.729  | 33.99 | 26.06  |
> | Ours    | **142.965** | **24.008** | **35.08** | **28.64**  |

---

> > ### Comment · Reviewer_1nz3 · 2025-08-01
> >
> > Thank you for the response. I have read the rebuttal and the other reviewers’ comments, and I will keep my score as 5. However, I believe that including comparisons on the full Aesthetic-4K dataset (instead of only the evaluation subset) in the final version would further strengthen the paper.

---

> > > ### Author Response · Authors · 2025-08-05
> > >
> > > We are glad to receive your feedback! Thank you once again for your insightful and constructive comments. We assure you that we will include a comparison with the full Aesthetic-4K dataset in the revised version.

---

### Official Review · Reviewer_WG7N · 2025-06-29

**Clarity:** 3
**Significance:** 2
**Originality:** 2
**Rating:** 4
**Confidence:** 4

**Summary:**

The paper tackles key limitations in ultra-high-resolution (UHR) text-to-image generation by introducing both a novel dataset and a post-training method for diffusion models. First, the authors release a large-scale dataset of UHR images paired with rich, detailed captions, designed to better support training and evaluation at high resolutions. They then propose a post-training technique to enhance diffusion model performance specifically in the UHR regime. Finally, the authors introduce a new benchmark built on their dataset and use it to assess the effectiveness of their method. The paper includes both quantitative and qualitative comparisons with state-of-the-art methods as well as an ablation study to evaluate the contributions of different components of their approach.

**Questions:**

1. Could you please specify the license under which the images in your dataset are released? It is essential to confirm that all images are sourced under permissive licenses that allow redistribution and use for the purposes outlined in this paper. If you are unable to provide clear licensing information, it may not be appropriate to consider the dataset and benchmark as formal contributions of this work.
2. You employ the LAION aesthetic quality predictor to filter out images below a certain threshold. Given the subjective nature of aesthetic quality, could you discuss potential biases this introduces into the dataset? How might these biases propagate into downstream models trained using your dataset or benchmark? Additionally, what are the potential societal impacts, particularly concerning aesthetic preferences or representational diversity?
3. The current discussion of limitations is restricted to the post-training method. Could you expand this to also address the limitations and downstream impacts of the dataset and benchmark, especially in terms of potential accessibility limitations, representational imbalances, or ethical concerns? This section would benefit from a deeper reflection on social implications and possible future extensions.
4. Could you explicitly state the architecture and configuration for Model A used in the ablation study (Table 4) and include it in Table 3 for a complete comparative evaluation? This will support replicability and help contextualize performance improvements. Additionally, please discuss the negative results more thoroughly. For example, in Section 5.3, it is clear that some methods underperform on CLIP scores. Understanding these failure modes would strengthen the paper’s empirical analysis and provide valuable insights for future work..

To improve the quality score: Include Model A in Table 3 and provide a clear analysis of how it compares to other models. Expand your discussion to include both positive and negative results, providing explanations for observed limitations or failure modes.

To improve the clarity score: Clearly state the model architecture and configuration for Model A to ensure replicability. Also clarify how evaluation dataset splits are selected.

To improve the significance score: Provide explicit licensing information for your dataset and discuss the biases and limitations of your methods in more depth, particularly those associated with data filtering and curation practices.

To improve the originality score: Provide explicit licensing information for your dataset and contextualize your post-training method by comparing it to other post-training or fine-tuning techniques, including from adjacent domains. This would help clarify what is fundamentally new about your approach.

Final Note: This is a promising and well-executed paper with strong potential contributions. However, confirmation of dataset licensing is essential. If the licensing is permissive and properly addressed, I would be happy to revisit the overall evaluation positively, as the work is otherwise of high technical quality and broad interest to the NeurIPS community.

**Ethical Concerns:**

["Major Concern: Data privacy, copyright, and consent"]

**Final Justification:**

The authors provided important clarifications that improved the technical quality and clarity of the paper.

Most significantly, they addressed the licensing concerns regarding their dataset and acknowledged that not specifying the license is unacceptable. My initial recommended score was based on the assumption that their failure to specify the license of the source material for the dataset would constitute an ethics violation and prevent their dataset from being considered within their contributions. However, their clarification remedied that, and I'm willing to adjust the recommended score upwards to factor in the dataset which improves the overall significance and originality of the paper.

However, I'd like to caveat that this is contingent on their response to the licensing concerns being valid; I'm not an expert in this area, so I'm only improving my recommended score based on my reading of their response. The ethics reviewer should hold the final say on whether or not their response is adequate and thus decide if the paper is free of ethics violations.

**Limitations:**

No. As mentioned above:
- Given the increasing scrutiny and societal concerns around synthetic media, the authors should minimally acknowledge ethical and societal risks associated with enabling more photorealistic high-resolution image generation. A more complete analysis would enhance the credibility and responsibility of the work.
- Overall, the discussion of limitations is notably insufficient. While the paper introduces three major contributions (dataset, benchmark, and post-training method), only the training method’s limitations are addressed. The dataset and benchmark also warrant discussion—particularly around dataset curation, representation, biases, and potential misuse, which are essential considerations for any work involving generative AI.

**Quality:**

3

**Strengths And Weaknesses:**

Quality

Strengths
- The methodology for constructing the dataset is carefully considered and well-documented. The filtering and captioning strategies demonstrate a rigorous approach to dataset curation.
- The proposed post-training method is well-motivated and technically sound, with strong rationale provided for the introduced components.
- The experimental design is robust and comprehensive. It includes a solid set of baselines, a well-structured ablation study, and both quantitative and qualitative evaluations which contribute to a well-rounded empirical assessment of the proposed approach.

Weaknesses
- In Table 1 and Lines 144 to 148, the paper notes the use of the LAION Aesthetics predictor to filter out images that do not meet an aesthetic threshold. However, the implications of this filtering are not discussed. Given concerns around aesthetic filtering (e.g., reinforcing biases or discarding content types disproportionately), the authors should reflect on whether this introduces systematic limitations or skews in the dataset and what the implications are if any. The shortcomings of LAION might be worth discussing here as well.
- Lines 153 and 154 mention that 2000 images were selected for the evaluation set, but no explanation is given for the selection strategy. It is important to clarify whether this set was sampled randomly or curated for diversity, and whether any balancing criteria were used. Additionally, breaking down the distributions in Figure 3 by train/evaluation split could strengthen the claim of unbiasedness and representativeness in the evaluation set.
- In the quantitative evaluation section from Lines 252 to 259, the paper highlights strong FID patch results for the proposed method but neglects to comment on its comparatively weak IS patch performance. The authors should discuss negative results as well for a more balanced analysis, especially since the IS patch to FID patch performance discrepancy is stark and merits analysis. Acknowledging and exploring possible reasons behind this could help surface limitations of the method and improve the robustness of the evaluation.
- In Section 5.3, negative or inconclusive findings from the ablation study (specifically, the performance drop in CLIP score when DOTS and SWFR are used, as seen in Model D vs. Model A) are not discussed. These results suggest that the proposed training strategy does not universally improve all evaluation metrics, and discussing them would lead to a more balanced and complete picture of the method’s strengths and trade-offs.
- For better alignment between the ablation study and the main comparison in Table 3, Model A of the ablation study should be included in Table 3. Without it, it is difficult to isolate how much of the observed performance improvement comes from the proposed training method versus the base model choice. This is especially relevant given that Model A already outperforms all models in Table 3 on CLIP and is the second-best performer FID patch.

Clarity

Strengths
- The introduction is well-written, with a well-defined problem statement and insightful contextualization within existing work. It effectively sets the stage for the paper’s contributions.
- The paper's structure is logical and easy to follow. Technical concepts are explained with a high degree of precision and care. The motivations behind each design choice are clearly conveyed, contributing to strong overall readability.
- The visual presentation of the dataset through examples and image illustrations is helpful in understanding the scale and diversity of the data, though some could be more space-efficient if space is a constraint. The dataset comparison table in Table 2 helps contextualize the proposed dataset relative to existing datasets.

Weaknesses
- The pipeline presented in Table 1 is functionally informative, but tables are not an effective way to convey procedural flows. Consider replacing this table with a pipeline diagram, which would more intuitively communicate the processing steps and their relationships.
- In Lines 81 to 85, it would be clearer to explicitly list the three main contributions: (1) a UHR image-caption dataset; (2) a post-training strategy for diffusion models; and (3) a benchmark for UHR text-to-image generation. This would clarify the paper’s scope and focus early on.
- Typo in Line 144: the term should be “Aesthetic” (not “Aesthetics”).
- Line 141: Add a citation or footnote for Shannon entropy, as it’s a referenced concept and not common knowledge to all readers.
- Figure 3 appears too early in the paper; it should be moved to Section 3, where the dataset is formally introduced, for better alignment between figures and content.
- Lines 165 to 170 repeat points made earlier in the paper. Consider either integrating this material into the earlier sections (e.g., Introduction or Dataset description) or trimming it for conciseness.
- In Section 5.3, the underlying model architecture and configuration used for Model A should be clearly specified. Since its scores in Table 4 do not align with any models in Table 3, it is unclear what base model was used. Providing this information is necessary to enable replication of the results and to properly contextualize the improvements shown in the ablation study.

Significance

Strengths
- The proposed UHR dataset, if confirmed to be publicly usable (pending clarification on image licensing), has the potential to serve as a valuable resource for the community. High-quality, caption-rich ultra-high-resolution image datasets remain scarce, and this contribution could help advance research in text-to-image generation at scale.
- The training strategy introduced in the paper demonstrates promise, with empirical results suggesting improved performance across several metrics. If validated more broadly, this could become a useful tool for practitioners working with diffusion models in the UHR regime.
- The UltraHR-eval4K benchmark, derived from the proposed dataset, adds further utility and could help standardize evaluation practices in UHR image synthesis tasks.

Weaknesses
- The significance of the proposed training method is primarily evaluated on the authors’ own benchmark. A more convincing demonstration of significance would involve testing the generalizability of the proposed method—i.e., evaluating models trained with the new strategy on existing, established benchmarks for text-to-image generation.
- Given the increasing scrutiny and societal concerns around synthetic media, the authors should minimally acknowledge ethical and societal risks associated with enabling more photorealistic high-resolution image generation. A more complete analysis would enhance the credibility and responsibility of the work.
- Overall, the discussion of limitations is notably insufficient. While the paper introduces three major contributions (dataset, benchmark, and post-training method), only the training method’s limitations are addressed. The dataset and benchmark also warrant discussion—particularly around dataset curation, representation, biases, and potential misuse, which are essential considerations for any work involving generative AI.

Originality

Strengths
- The proposed dataset is novel in scale, resolution, and quality, and stands out for its rigorous collection methodology as presented in Table 2 and detailed in Section 3. Pending clarification on image licensing, the dataset can contribute meaningfully to the field.
- The post-training method introduced offers a relatively new approach to improving text-to-image generation in the ultra-high-resolution regime.
- The related work discussion in Section 2 is comprehensive and does a good job of situating the paper within the landscape of image generation and large-scale diffusion training. The comparative framing helps clarify how the proposed contributions differ from or extend prior work, underscoring their originality.

Weaknesses
- The paper would benefit from a more explicit comparison to existing post-training or fine-tuning strategies for diffusion models (even from adjacent domains which apply a similar approach) to better contextualize the novelty of the proposed method. Identifying and comparing the proposed approach to similar strategies would help underscore how the proposed approach builds upon or diverges from established ideas.

---

> ### Author Rebuttal · Authors · 2025-07-29
>
> ## Quality
> 1) We appreciate the reviewer’s valuable comments on aesthetic filtering. 1) Existing datasets, including LAION, OpenVid-1M, OpenHumanVid, and OmniStyle-1M, commonly employ aesthetic filtering to enhance visual quality by filtering out low-quality samples. Our filtering strategy aligns with these practices, aiming to ensure a cleaner training corpus. Notably, to prevent overly aggressive filtering, we removed only the bottom 50% of images based on aesthetic scores. This approach serves as a lightweight filter, primarily targeting severely degraded or visually corrupted content. 2) We recognize that aesthetic filtering may introduce biases by excluding images with monochrome or low contrast, text-heavy or document-like images, and non-photographic compositions. Although these may seem less “aesthetic,” they often contain important semantic information for text-to-image tasks, so filtering them reduces content diversity. 3) The LAION Aesthetics Predictor is trained on subjective aesthetic scores and may inherently reflect cultural or stylistic biases, favoring certain photographic conventions. It tends to exhibit reduced tolerance for minimalistic, abstract, or utilitarian designs (e.g., charts, educational diagrams, UI mockups). Such biases can inadvertently lead to the exclusion of valuable non-photographic content.
>
> 2) The UltraHR-eval4K evaluation set was randomly sampled from the entire dataset under the constraint that both height and width exceed 4096 pixels. This ensures that the evaluation set remains unbiased and reflects the natural data distribution within the full dataset. As shown in **Table 1**, we provide statistical insights for our UltraHR-eval4K benchmark. We will include these statistics in the revised version of the paper
>
> 3) FID measures the distance between two data distributions. Our model achieves strong performance on FID_patch, indicating that the generated image patches exhibit high structural and textural similarity to real high-quality images, reflecting the effectiveness of our approach in preserving fine-grained local details. However, the IS_patch score is relatively low, which warrants further analysis. We attribute this disparity primarily to the characteristics and inherent limitations of the IS. IS evaluates the quality and diversity of generated images via a pre-trained classifier. In the context of UHR image generation, evaluating local patches using IS presents two key challenges: (1) UHR image patches often lack clear semantic category attributes, which reduces the classifier's confidence and thus lowers IS scores; and (2) the classifier used in IS is typically trained on low-resolution data, which introduces a significant distribution gap when applied to high-resolution content—further undermining its reliability. Moreover, since IS is based on classification confidence, it is not well-suited to assess the local detail quality of UHR images. As further evidence, as shown in **Table 2**, Model A—trained solely on our UHR dataset—already yields a low IS_patch score. This strongly suggests that the discrepancy stems more from the characteristics of the data than from the design of our method.
>
> 4) We sincerely apologize for not providing a more detailed discussion of this issue in the paper. In the ablation study, the observed drop in CLIP score when comparing Model D to Model A reflects a limitation of our frequency-aware post-training strategy, which incorporates both DOTS and SWFR. This strategy primarily focuses on enhancing detail learning by allocating more training capacity to the denoising stages in the later sampling steps. However, this reallocation inevitably reduces the emphasis on the earlier denoising steps, which are more critical for modeling global semantics. As a result, the model’s ability to generate semantically rich and globally coherent content may be compromised, thereby affecting the CLIP score.
>
> 5) As shown in **Table 2**, we have already included Model A in the main comparison. Model A refers to the variant trained solely on our proposed dataset using full fine-tuning. The experimental results demonstrate that our curated dataset alone can substantially enhance the generative capability of the baseline model. To ensure a fair comparison, the training setup, hardware environment, and number of training iterations are kept consistent with those of Model D.
>
> ---
>
> ## Clarity
> 1) We will replace the current table with a pipeline figure to present the data processing steps in a clearer and more objective manner.
>
> 2) In the revised paper, we will explicitly state the three key contributions to clarify the scope and focus of the paper.
>
> 3) We will correct “Aesthetics” to “Aesthetic” in Line 144 in the revised version.
>
> 4) We will add an appropriate citation or footnote for Shannon entropy **[1]** in Line 141 to ensure clarity and proper attribution for readers unfamiliar with the concept.
>
> 5) We agree that Figure 3 would be better placed in Section 3. We will revise the paper accordingly to improve the alignment between figures and corresponding content.
>
> 6) We will review Lines 165–170 and either incorporate key points into earlier sections or remove repetition to improve clarity and flow.
>
> 7) Model A shares the same architecture as our full model D, both based on SANA. Specifically, Model A refers to the variant trained solely on our proposed dataset using full fine-tuning. To ensure a fair comparison, the training setup, hardware environment, and number of training iterations are kept consistent with those of Model D.
>
> [1] C. E. Shannon, "A mathematical theory of communication," in The Bell System Technical Journal.
>
> ---
>
> ## Significance
> 1) We conduct a quantitative comparison on the publicly available Aesthetic-Eval@4096 benchmark, as reported in **Table 3**. This evaluation set contains 195 image-text pairs, where all images have a short side greater than 4096 pixels. The results clearly demonstrate the superior performance of our method, supporting its robustness and generalizability beyond our proposed benchmark.
>
> 2) We acknowledge that advances in photorealistic high-resolution image generation raise important ethical and societal concerns, including risks related to misinformation, privacy, and misuse of synthetic media. We emphasize the need for increased awareness of these issues and encourage the community to collaboratively develop ethical guidelines and mitigation strategies. We will include a brief discussion of these ethical challenges in the revised manuscript.
>
> 3) The main limitation of our dataset currently lies in the relatively small amount of portrait data, which results in limited improvements over prior methods in UHR portrait generation, as also illustrated in Figure 8 of the paper. In future work, we plan to collect more high-quality UHR portrait images to enhance our model’s performance specifically in UHR portrait synthesis.
>
> ---
>
> ## Originality
>
> 1)  We have added a comparison experiment with LoRA, as shown in **Table 4**, to further highlight the effectiveness of our proposed frequency-aware post-training strategy. Compared to existing post-training approach, our method demonstrates particularly notable improvements in detail enhancement, as reflected by the FID_patch metric.
>
> ---
>
> ## Questions
> 1) We confirm that our dataset will be released under the Creative Commons Attribution-NonCommercial (CC BY-NC) license, which permits redistribution and use for non-commercial purposes, such as academic research. This license ensures a permissive usage framework while maintaining responsible use boundaries. We are fully committed to open science—upon acceptance of the paper, we will release all data, code, and trained models to the community. The primary goal of our dataset is to support and enrich open research in the field of ultra-high-resolution text-to-image generation.
>
> 2) For aesthetic, please refer to our response to Comment 1 in the Quality section.
>
> 3) For limitations, please refer to our response to Comment 3 in the Significance section. Furthermore, we have discussed the broader social implications and ethical considerations in our response to Comment 2 in the Significance section.
>
> 4) Please refer to Comments 3 and 4 in the Quality section for a detailed discussion and clarification regarding Model A.
>
> ---
>
> Table 1. (a) Distribution statistics of image resolutions in the UltraHR-eval4K benchmark.
> | Resolution  | 0k-4k  | 4k-5k   | 5k-6k   | 6k-7k   | 7k-8k   |  8k-   |
> |---------|--------|--------|--------|--------|--------|--------|
> | Quantity|  0 |  69 |  887 | 740 |  233  |  71  |
>
> (b) Caption length distribution in our UltraHR-eval4K benchmark.
> | Length  | 0-60  | 60-100   | 100-120   | 120-140   | 140-   |
> |---------|--------|--------|--------|--------|--------|
> | Quantity|  7 |  349  |  1073 | 521 |  50  |
>
> ---
>
> Table 2. Quantitative comparison on  our UltraHR-eval4K (4096 × 4096) benchmark.
> | Method  | FID   | FID_patch   | IS   | IS_patch   | CLIP   | FG-CLIP   |
> |---------|--------|--------|--------|--------|--------|--------|
> | SANA|   37.070|  38.795 | 11.778|  **5.649** |  31.70|  28.60|
> | Model A | 33.995 | 20.932 | 12.502 | 5.020  | **31.85** | 28.65  |
> | Ours (Model D)   | **31.748** | **15.795** | **12.995** | 5.104  | 31.82  | **28.68**  |
>
> ---
>
> Table 3. Quantitative comparison on Aesthetic-Eval@4096.
> | Method  |    FID     | FID_patch    | CLIP  | FG-CLIP |
> |-------------|-------------|-------------|---------------|-------------------|
> | Pixart-σ    | 150.593 | 44.702  | 34.88 | 28.48  |
> | SANA    | 146.027 | 37.031  | 34.62 | 28.61  |
> | DIFF4K  | 152.790 | 39.729  | 33.99 | 26.06  |
> | Ours    | **142.965** | **24.008** | **35.08** | **28.64**  |
>
> ---
>
> Table 4. Comparison experiment with LoRA.
> | Method  |    FID     | FID_patch    | CLIP  |
> |-------------|-------------|-------------|---------------|
> | LoRA  |  35.07 | 35.02  | 31.80 |
> | Ours    | **31.74** | **15.79** | **31.82** |

---

> > ### Comment · Reviewer_WG7N · 2025-08-06
> >
> > Thank you for the thoughtful rebuttal. I’m glad to hear that some of the feedback has been helpful and will be incorporated. Please find my follow-up responses and clarifications below:
> >
> > 	•	Regarding Quality Point 1:
> > The additional details you provided are valuable and should ideally be included in the final paper. I also recommend ensuring that the concerns raised by the Ethics reviewer are addressed explicitly in this section as well.
> > 	•	Regarding Quality Point 2:
> > While random sampling is common, it does not guarantee unbiased coverage. For example, if your dataset consists mostly of cat images and only a few dog images, random sampling could result in an evaluation set with no representation of certain classes. I strongly recommend a more principled sampling approach (such as stratified sampling) that considers the class distribution to ensure better representativeness.
> > 	•	Regarding Quality Points 3 & 4:
> > The insights you’ve shared in the rebuttal are helpful, but they should be reflected in the paper. Not discussing negative or contradictory results weakens the paper and gives the impression of selective reporting. Including this analysis will improve the technical depth and credibility of your study.
> > 	•	Regarding Quality Point 5:
> > This clarification is useful and should be incorporated into your main experimental section. Discussing whether your method had a meaningful impact is important for providing a complete and transparent evaluation.
> > 	•	Regarding Clarity Point 5:
> > This is not clearly communicated in the current draft. Section 5.3 introduces Model A without specifying its architecture. You should explicitly state what model is used rather than expecting readers to infer this from the similarity of scores with other models in Table 3.
> > 	•	Regarding Significance Point 1:
> > I strongly encourage including results from additional benchmarks to demonstrate generalizability. Without this, the impact of your method remains limited to your custom benchmark, which restricts its broader relevance.
> > 	•	Regarding Originality Point 1:
> > Similarly, incorporating comparative analysis or discussion with other post-training or frequency-domain methods would better contextualize your work and clarify its novelty. This will help readers understand how your approach differs from or improves upon prior work.
> > 	•	Regarding Question 1 (Licensing and Dataset Release):
> > There is a critical distinction between applying a Creative Commons license to your dataset and ensuring that the underlying source material is compatible with that license. Your rebuttal to the Ethics reviewer’s concerns is still somewhat ambiguous. A statement that “most” images are CC-licensed is insufficient. Unless you can guarantee that all images are permissively licensed, releasing the dataset under a CC license may violate the terms of individual image sources. Please ensure this is addressed unambiguously and in full compliance with licensing norms.

---

> > > ### Author Response · Authors · 2025-08-06
> > > **Respond to Reviewer WG7N**
> > >
> > > Thank you very much for your valuable feedback, which has greatly enhanced the quality of our paper.
> > >
> > > **Regarding Quality Points 1, 3, 4, and 5**, we sincerely assure you that the additional experiments and analyses included in the rebuttal will be incorporated into the final paper. These additions will further strengthen the technical depth and credibility of our study.
> > >
> > > **Regarding Quality Points  2 and Significance Point 1**, we fully understand your concerns about the random sampling used in constructing our benchmark. We recognize the potential bias it may introduce. However, re-creating the benchmark requires significant time, and it would be challenging for us to present new results during the discussion phase. Nevertheless, the random sampling still provides value in ensuring the reliability of overall distribution. For  the future work, we plan to adopt stratified sampling to carefully consider class distributions, ensuring better representativeness. Furthermore, we have conducted a quantitative comparison on the publicly available **Aesthetic-Eval@4096 benchmark [1]**, as shown in **Table 1**. The results clearly highlight the superior performance of our method, demonstrating its robustness and generalizability.
> > >
> > > **Regarding Clarity Point 5**, we would like to clarify that Model A shares the same architecture as our full model D, both based on **SANA**. Specifically, Model A refers to the variant trained solely on our proposed dataset using full fine-tuning. To ensure a fair comparison, the training setup, hardware environment, and number of training iterations are kept consistent with those of Model D.
> > >
> > > **Regarding Originality Point 1**, we have added a comparison experiment with **LoRA**, as shown in **Table 2**, to further highlight the effectiveness of our proposed method. Compared to existing methods, our approach demonstrates particularly notable improvements in detail enhancement, as reflected by the FID_patch metric.
> > >
> > > **Regarding Question 1 (Licensing and Dataset Release)**, as we mentioned in our response to the Ethics reviewer, the **majority** of the images in our dataset were sourced from public websites under permissive licenses. **Additionally**, a **small portion** of the dataset contains personally captured photographs, for which usage permission has been obtained. Given this data sourcing, we are confident that our constructed dataset, UltraHR-100K, is in full compliance with copyright laws and privacy regulations. **Therefore, we are certain that our dataset fully adheres to licensing norms.**
> > >
> > > ---
> > >
> > > Table 1. Quantitative comparison on **Aesthetic-Eval@4096 [1]**.
> > > | Method      | FID    | FID_patch | CLIP  | FG-CLIP |
> > > |-------------|--------|-----------|-------|---------|
> > > | Pixart-σ    | 150.593 | 44.702    | 34.88 | 28.48   |
> > > | SANA        | 146.027 | 37.031    | 34.62 | 28.61   |
> > > | DIFF4K      | 152.790 | 39.729    | 33.99 | 26.06   |
> > > | Ours        | **142.965** | **24.008**    | **35.08** | **28.64**   |
> > >
> > > ---
> > > Table 2. Comparison experiment with LoRA.
> > > | Method | FID   | FID_patch | CLIP  |
> > > |--------|-------|-----------|-------|
> > > | LoRA   | 35.07 | 35.02     | 31.80 |
> > > | Ours   | **31.74** | **15.79**     | **31.82** |
> > >
> > > ---
> > > [1] Diffusion-4k: Ultra-high-resolution image synthesis with latent diffusion models. Proceedings of the Computer Vision and Pattern Recognition Conference. 2025.

---

> > > > ### Comment · Reviewer_WG7N · 2025-08-06
> > > >
> > > > Thank you again for the rebuttal. I have a few clarifications and follow-up comments on key points:
> > > >
> > > > Regarding Quality Point 2 and Significance Point 1 (Random Sampling):
> > > > I disagree with the assertion that “random sampling still provides value in ensuring the reliability of overall distribution.” Random sampling, by its very nature, introduces variance and does not guarantee class balance or representativeness, especially in the absence of stratification or other controls. I would recommend that the paper acknowledge this limitation explicitly, rather than suggest that random sampling ensures distributional reliability.
> > > >
> > > > Additionally, the reference to Aesthetic-Eval@4096 is not persuasive in this context. Generalization performance alone is not sufficient evidence of dataset impartiality or robustness. A proper analysis of class or content distribution (e.g. aesthetic categories, subject matter diversity, source bias) would be needed to make any such claims credible.
> > > >
> > > > Regarding Clarity Point 5 (Model A):
> > > > Thank you for the clarification. However, I would urge you to explicitly describe Model A in the paper itself, as its introduction is currently abrupt and lacks context. Without a description of the architecture or configuration, readers are left confused. This can easily be resolved through a brief sentence clarifying the model’s origin and setup.
> > > >
> > > >
> > > > Regarding Question 1 (Licensing and Dataset Release):
> > > > Apologies if my point comes across as overly pedantic, but I do think the current answer remains ambiguous. Phrases such as majority were this or a small proportion were that feel vague and evasive. Given that the dataset was web-scraped, the burden of proof regarding licensing is higher due to the inherent risk of including copyrighted or restrictively licensed images.
> > > >
> > > > I strongly recommend providing a clear breakdown of the data sources, associated licenses, and evidence that 100% of the images are covered by permissive licenses. This level of transparency is essential for NeurIPS and would allow the dataset contribution to be fairly and confidently considered as part of your paper’s core claims.
> > > >
> > > > Overall Assessment:
> > > > If the above issues are properly addressed and documented, I would be willing to revisit and improve my overall score.
> > > >
> > > > I would also like to echo a point made by reviewer UXyJ: your work attempts to cover both the creation of a large-scale dataset and benchmark, as well as a novel post-training method. These are substantial contributions in their own right, but the breadth may have constrained the depth of analysis in certain areas. You may find greater impact and clarity by splitting these into two separate submissions in the future (e.g., a main track methods paper and a datasets/benchmarks paper). Each would be stronger with more focused writing, dedicated analysis, and clearer framing of contributions.

---

> > > > > ### Author Response · Authors · 2025-08-06
> > > > > **Respond to Reviewer WG7N**
> > > > >
> > > > > Thank you once again for your response. Your feedback has been extremely helpful in guiding us to *further refine and expand* our work.
> > > > >
> > > > > **Regarding Quality Point 2 and Significance Point 1 (Random Sampling):** We apologize for any misunderstanding regarding the reliability of the random sampling. We fully acknowledge that random sampling inherently introduces variance and does not guarantee class balance or representativeness, which represents a limitation of the current benchmark. We will revise the limitations of our work in the final paper, including the dataset, method, and benchmark. In future extensions of this work, we will further address these issues.
> > > > >
> > > > > **Regarding Clarity Point 5 (Model A):** We apologize for the confusion caused by the lack of detailed configuration information regarding Model A. We will include a clear and explicit description of Model A’s configuration and setup in the final paper to eliminate any uncertainty.
> > > > >
> > > > > **Regarding Question 1 (Licensing and Dataset Release):**
> > > > > We apologize for the vagueness in our previous response. We would like to clarify that the web-scraped images in our data sources were sourced from three free websites:
> > > > > Unsplash, Pexels, and Mediastorm (YSJF).
> > > > >
> > > > > For Unsplash, the licensing details are as follows: *"Unsplash visuals are made to be used freely. Our license reflects that. All images can be downloaded and used for free, for both commercial and non-commercial purposes. No permission is needed (though attribution is appreciated!)"*. This information is sourced from the Unsplash website.
> > > > >
> > > > > For Pexels, the licensing details are as follows: *"All photos and videos on Pexels can be downloaded and used for free. Attribution is not required. Giving credit to the photographer or Pexels is not necessary but always appreciated. You can modify the photos and videos from Pexels. Be creative and edit them as you like."* This information is sourced from the Pexels website.
> > > > >
> > > > > For Mediastorm (YSJF), we regret that the website does not have an explicit license declaration. However, we want to clarify that the images on Mediastorm are produced and captured by their own team. Although there is no formal license declaration, the website clearly outlines three usage models: free personal use, free personal commercial use, and free commercial use for enterprises. Thus, the usage rights are highly permissive.
> > > > >
> > > > > In conclusion, we can confidently state that the web-scraped images are 100% covered by permissive licenses. Additionally, some images in our dataset were personally photographed by our team members, with usage rights and permissions determined by our research group, and these also adhere to permissive licenses. Therefore, we are certain that our dataset fully complies with licensing norms.
> > > > >
> > > > > We apologize for the inconvenience, but due to NeurIPS regulations, we are unable to directly provide the license links to the websites. If you have any further questions, please feel free to ask, and we will do our best to address and resolve them.

---

> > > > > > ### Comment · Reviewer_WG7N · 2025-08-07
> > > > > >
> > > > > > Thank you for your detailed clarifications.
> > > > > >
> > > > > > Given the additional explanations regarding the experimental methodology and the expanded discussion of quality and significance, I am willing to raise my overall score to a borderline accept. I do still believe that the paper would benefit from greater focus: covering both the dataset and the post-training method within a single paper naturally limits the depth of discussion and impact for either under the page constraints. Future versions of this work might benefit from splitting these into distinct contributions.
> > > > > >
> > > > > > Regarding licensing:
> > > > > > Your latest response is more satisfactory, and I appreciate the detailed breakdown. That said, the licensing situation for the Mediastorm (YSJF) portion of the dataset remains somewhat unclear to me. Nonetheless, I will proceed under the assumption that the licensing concerns have been adequately addressed and that the dataset can be considered a valid contribution in the context of this submission. However, I strongly recommend that you include the complete licensing breakdown as a comment for the ethics reviewer, who will be in a better position to evaluate and formally assess the dataset’s compliance with NeurIPS’ policies on data usage and release.
> > > > > >
> > > > > > Thank you again for engaging constructively with the reviewer feedback.

---

> > > > > > > ### Comment · Reviewer_WG7N · 2025-08-07
> > > > > > >
> > > > > > > Additionally, I strongly suggest that in your final submission or your public dataset repo, you include very clear licensing information for all source material. Your team has done good work - and it would be a waste for it to be hampered by licensing issues. While the Unsplash license is well-known, I'm not sure if the Pexels and Mediastorm "licenses" you've described will cut it.

---

> > > > > > > > ### Author Response · Authors · 2025-08-07
> > > > > > > > **Respond to Reviewer WG7N**
> > > > > > > >
> > > > > > > > We sincerely appreciate your active engagement and the high regard you have shown for our work.
> > > > > > > >
> > > > > > > > [1] Thank you for your valuable suggestions regarding aesthetic bias, experimental analysis, experimental setup, ethical considerations, paper layout and formatting, and the licensing of our dataset. These discussions have greatly enhanced the quality of our research. We assure you that we will revise the final paper to incorporate all of these discussions, as well as the updates you recommended, to improve the overall quality of the paper.
> > > > > > > >
> > > > > > > > [2] We are also deeply grateful for your insightful perspectives on the current limitations and future extensions of our work. We fully agree that splitting the dataset and the method into distinct contributions, with in-depth discussions on each, will significantly strengthen the quality of the research. Your insights not only have been invaluable to the current work but also will have a lasting impact on future extensions.
> > > > > > > >
> > > > > > > > Regarding Mediastorm (ysjf), it is the official media library of the renowned content creation team, Mediastorm. The platform aims to provide a professional, high-quality, and creator-friendly resource hub. All materials on the site are available for free, including for commercial projects. Further details can be found on their website. Additionally, we assure you that we will include clear licensing information for all source materials in both the final submission and the public dataset repository.
> > > > > > > >
> > > > > > > > Finally, we sincerely thank you for considering an adjustment to your final evaluation. We greatly appreciate your time and participation.

---

> > > > > ### Author Response · Authors · 2025-08-07
> > > > > **We sincerely appreciate Reviewer WG7N**
> > > > >
> > > > > Once again, we sincerely appreciate your **thoughtful, comprehensive, and detailed** comments. Your feedback has strengthened our paper and research in numerous ways, and we are truly grateful for you.
> > > > >
> > > > > We would like to clarify that the focus of our current paper is to enhance the capabilities of pre-trained T2I models in generating UHR images by constructing a high-quality **dataset** and a detail-guided **method**. As such, building a high-quality evaluation benchmark is not the primary focus of this work. We plan to address this aspect in future work.
> > > > >
> > > > > Finally, if you have any further questions or concerns, please do not hesitate to let us know. We will do our utmost to address and resolve them.
> > > > >
> > > > > Thank you again for your invaluable feedback.

---

> ### Author Response · Authors · 2025-08-05
> **Respond to Reviewer WG7N**
>
> Dear Reviewer Reviewer WG7N,
>
> As the discussion deadline approaches [Aug 06 '25 (AoE)], we fully understand that you may have other pressing commitments. We are genuinely grateful for the time and effort you have dedicated to reviewing our submission, as well as for your insightful and constructive feedback. The discussion process not only provides an invaluable opportunity to improve the quality of the paper for the authors, but also deepens the reviewers' understanding of the work, contributing to the overall advancement of the academic community. We sincerely hope to continue addressing any unresolved concerns through our responses. Thank you once again for your invaluable time and for your role in helping us refine and strengthen our work!

---

### Official Review · Reviewer_4Jyg · 2025-07-01

**Clarity:** 4
**Significance:** 2
**Originality:** 2
**Rating:** 4
**Confidence:** 4

**Summary:**

The paper tackles the problem of high-quality, detail-rich image generation, by proposing a new dataset UHR100K for training. UHR100K is collected and filtered with well-designed strategies.
In addition to that, a frequency-aware post-training method is proposed for detail synthesis.
With the new UHR100K dataset and the new training strategy, the authors improve the synthesis capability of pre-trained T2I models.

**Questions:**

- Why the eval set "UltraHR-eval4K" is named after "4K" while it actually only consists of 2K images (Table 2)?
- What is $\lambda_{freq}$ used in your experiments?

**Ethical Concerns:**

["NO or VERY MINOR ethics concerns only"]

**Final Justification:**

The papers comes with a new large-scale dataset. The authors have addressed all of my concerns in the rebuttal. I think the paper is ready for acceptance.

**Limitations:**

No limitations are discussed. It is highly recommended for the authors to discuss.

**Paper Formatting Concerns:**

No concerns.

**Quality:**

3

**Strengths And Weaknesses:**

The paper shows a straight-forward post-training pipeline for enhancing the detail synthesis for T2I models. The simple idea shows promising results. However, there are still issues with the detail richness and color saturation in the results. Also, the balance between "complicate textures" and "simplicity" needs further discussions.

My current rating for the paper is Borderline Accept.

## Strengths
### Idea/Motivation
- The motivation of proposing UHR100K is straight-forward and intuitive. Instead of using complex training-free methods, which still struggle generating detail-rich images, the authors curate more and better data as a simple solution. a
- Helping models to capture the details in generation also makes sense. A new post-training strategy is proposed to compile the new dataset.

### Method
- The authors filter data based on content complexity, detail richness and aesthetic quality.
- Based on the observation that fine details are usually generated in the later denoising process, the authors propose to sample from a Beta distribution, whose emphasis is inclined to later timesteps.
- SWFR puts more focus on the high-frequency synthesis.

### Results/Experiments
- Table 3 shows the proposed dataset and SWFR brings a boost for multiple metrics.
- Ablation study in Table 4 shows the improvement for each component.

### Writing/Delivery
The paper is overall well-written with clear guidances for readers.

## Weaknesses
### Idea/Motivation
- While the capability to generate rich details is generally desirable, it sometimes may not align with all aesthetic preferences.
For example, Minimalism asks for simplicity and discourages the complexity in pictures. Will the proposed dataset and method introduce a bias towards rich visual contents? The paper might be more comprehensive if authors could discuss and study this bias, especially the differences before and after the post-training.

### Method
- The captions are generated from an LLM (Gemini 2.0). Are those captions aligned with human perceptions? Are longer captions mean better semantics?
- Following the last comment, other than Gemini, are any other LLMs suitable for captioning?

### Results/Experiments
- Unfortunately, although with more details, some results still show unrealistic styles, such as over-smooth or weird color tones. For example,  in the middle row of Fig. 4 (Supp.), there are no details on the skin and it is over-smoothed. Also on the page 8 in the supp., the color of "ours" looks unrealistic with a strong emphasis on the brown and yellow hues. Also on the page 11 in the supp., in my opinion the result from Diffusion-4K looks more natural.
- Although achieving the best performance on quantitative metrics in Table 3, it is more convincing if a user study is included, as most of the existing metrics still struggle aligning with human perceptions.
- There is no qualitative comparison between each component in the ablation study.
- It might be more convincing if the training strategy and the new dataset can be applied to more base models other than SANA.
- Given that this is a paper enhancing visual quality, it is highly recommended authors could include native images/outputs in the supplementary material as it is still far from the file limit (21.5MB v.s. 100MB). Some images shown in the supplementary material come with obvious compression loss (e.g., Fig. 11 in the supp.).

### Writing/Delivery
- L35-L36, "a open-source" -> "an open-source"
- In Fig. 5, the colors of legend is hard to distinguish. Some colors are similar (e.g., "$\alpha=0.5, \beta=0.5$" and "$\alpha=2, \beta=4$").

---

> ### Author Rebuttal · Authors · 2025-07-28
>
> We appreciate this valuable suggestion. However, due to this year’s NeurIPS rebuttal guidelines, we are unable to include visual results in the response. We will include the requested visualizations in the revised version of the paper, including qualitative examples for minimalism prompts and ablation study results.
>
> ---
> ## For the Idea / Motivation
> We thank the reviewer for raising the concern about potential bias toward rich visual content, which may not align with aesthetics such as minimalism. To investigate this, we conducted preliminary qualitative experiments using minimalist prompts before and after post-training. The results suggest that our method, while enhancing detail, does not significantly degrade the model’s ability to generate stylistically minimalist images. This robustness may stem from the strong generalization ability of the pretrained model.
> Our primary motivation is to improve model performance in Ultra-High-Resolution (UHR) scenarios, especially under complex content and fine-grained detail—areas where existing models often struggle. For simple or minimalist generation tasks, pretrained models such as FLUX or SD3 are often already sufficient without further UHR fine-tuning. We plan to include further discussion and illustrative minimalist examples in the revised version.
>
> ---
>
> ## For the Method
>
> To assess whether the long captions generated by Gemini 2.0 align with human perception, we conducted a user study, as shown in **Table 1**. Ten volunteers each evaluated 50 caption pairs (our original long captions vs. Gemini-generated short captions) based on semantic richness (the extent to which the caption captures semantic content of the image) and semantic accuracy (how accurately the caption reflects the image content), using a 1–5 scale (1 = very poor, 2 = poor, 3 = fair, 4 = good, 5 = excellent). We report the average scores across all participants. Additionally, we collected overall preference judgments between the two captions. Results show that the long captions are consistently rated higher in both semantic richness and accuracy, and are more frequently preferred, demonstrating their stronger semantic expressiveness. Additionally, in **Table 2**, we evaluated models using only the first sentence (short version) of the long captions. Results confirm that longer, richer captions lead to higher-quality image generation. While other LLMs like GPT-4o can also generate captions, we focused on Gemini due to current resource constraints. We plan to explore more LLMs in future work.
>
> ---
>
> ## Experiments
>
> ### 1. Unrealistic Styles
>
> We acknowledge that in some cases, our method produces over-smoothed textures or unnatural color tones. For skin smoothness, We believe this may be due to the scarcity of portrait data in our dataset, resulting in suboptimal portrait generation performance. This is a current limitation we plan to address by collecting more high-quality UHR portrait data. Unnatural color tones may also stem from base model limitations (SANA) and the difficulty of balancing detail enhancement with natural appearance. We will refine our post-training strategy to improve visual realism.
>
> ### 2. User Study
>
> As shown in **Table 3**, we conducted a user study with 5 volunteers evaluating 50 randomly selected cases. Images were rated on overall quality, detail quality, text-image alignment and preference. The results demonstrate the superiority of our method across all aspects.
>
> ### 3. Ablation Study
>
> We will include qualitative comparisons of ablation variants in the revised version.
>
> ### 4. Generalization to Other Models
>
> Due to limited computational resources, we currently focus on SANA. We plan to extend our training strategy and dataset to other models such as FLUX and SD3 in future work.
>
> ### 5. Image Compression
>
> All images were compressed using the same method due to Overleaf compilation limits. We apologize for not including original-resolution images and will release all data and outputs publicly after the paper accepted.
>
> ---
>
> ## Clarifications & Minor Issues
>
> - We apologize for the inconvenience caused by the grammatical and formatting issues. We will correct the grammatical error (“a open-source” → “an open-source”) and improve the color distinguishability in Figure 5.
> - The term **"4K"** in *UltraHR-eval4K* refers to **image resolution**, not the number of samples.
> - The frequency loss weight **$\lambda_{\text{freq}}$** is set to **1** in our experiments.
>
> ---
>
> ## Limitations & Future Work
>
> As mentioned, the lack of high-quality portrait data limits performance in portrait generation scenarios. Additionally, while our frequency-aware post-training strategy enhances detail synthesis, it slightly compromises text-image alignment, as evidenced by the CLIP score: Model A outperforms Model D in Table 4 of the paper. In future work, we will:
>
> - Collect more high-quality UHR portrait data.
> - Explore more effective training strategies to address this trade-off and improve robustness.
>
> ---
>
> Table 1. Caption User Study. Long_cap refers to our original long captions containing both global and fine-grained descriptions, while Short_cap refers to newly generated short captions that include only global descriptions.
> | Caption Type | Semantic Richness | Semantic Accuracy | Preference |
> |--------------|-------------------|-------------------|------------|
> | Short_cap    | 2.65              | 3.86              | 7%       |
> | Long_cap     | **4.87**              | **4.26**              | **93%**       |
>
> ---
>
> Table 2. Quantitative Comparison on UltraHR-eval4K Using Short Captions.  SANA-s and Ours-s denote results evaluated on the UltraHR-eval4K benchmark using short captions (the first sentence of the original long captions), while SANA and Ours indicate evaluation using the full original long captions.
> | Method   | FID     | FID_patch | IS     | IS_patch |
> |----------|---------|-----------|--------|----------|
> | SANA-s   |  41.234       |    39.657       |   11.216     |    5.394      |
> | SANA     | **37.070**  | **38.795**    | **11.778** | **5.649**    |
> ||
> | Ours-s   | 34.653  | 16.701    | 12.519 |  4.563   |
> | Ours     | **31.748**  | **15.795**    | **12.995** | **5.104**    |
>
> ---
>
> Table 3.  User study results conducted on our UltraHR-eval4K benchmark, evaluating overall quality, detail fidelity, text-image alignment and user preference across different methods.
> | Method   | Overall Quality | Detail Quality | Text-Image Alignment | Preference |
> |----------|-----------------|----------------|------------|-----------|
> | Pixart-σ   | 14% | 10% | 16% | 18% |
> | SANA     |   4%| 8% | 8% | 6% |
> | Diffusion4K   |   12% |4%	|6%|	6%|
> | Ours     |    **70%**|	**78%**| **72%** | **70%** |

---

### Note · Authors · 2025-08-12

We sincerely thank all the reviewers for their valuable feedback and are pleased to hear that they appreciated our work. Below is a summary of the positive comments provided by the reviewers:

Reviewer **4Jyg**: The motivation for proposing UHR100K is **straightforward and intuitive**. The idea of helping models capture details in generation is also **well-justified**. The paper is overall **well-written**, with clear guidance for readers.

Reviewer **WG7N**: The methodology for constructing the dataset is **carefully considered and well-documented**. The proposed post-training method is **well-motivated and technically sound**, with strong rationale provided for the introduced components. The introduction is **well-written**, with a clear problem statement and insightful contextualization within existing work. The proposed UHR dataset has the potential to serve as a **valuable resource** for the community.

Reviewer **1nz3**: The paper presents a rigorously curated UHR dataset (UltraHR-100K). The multi-dimensional filtering process is **clearly motivated** and ensures both diversity and fidelity of the data. Leveraging the proposed frequency-aware post-training strategy, the model achieves **strong performance** in UHR image synthesis.

Reviewer **UXyJ**: While I have limited confidence in this field, the overall work is **solid**.

We are pleased to hear that our response has addressed the questions and concerns raised by reviewers WG7N, 1nz3, and UXyJ. We would also like to extend our special thanks to Reviewer 1nz3 for maintaining the final score as "accept." Furthermore, we deeply appreciate the multiple discussions and insightful feedback from Reviewer WG7N, which greatly enhanced the integrity and depth of our work.

Finally, we would like to thank the Area Chair for their efforts in facilitating the discussion. We believe that our contributions have significant potential to inspire further exploration in the field of UHR image generation.

---

### Decision · Program_Chairs · 2025-09-17

**Decision:**

Accept (poster)

**Comment:**

This paper tries to enhance high-resolution text-to-image generation by proposing a new dataset and two techniques: Detail-Oriented Timestep Sampling (DOTS), to focus learning on detail-critical denoising steps, and Soft-Weighting Frequency Regularization (SWFR), to softly constrain frequency components. All reviewers are on the positive side, in particular acknowledging the importance of the dataset. The authors believe that they can resolve concerns regarding ethical reviews, and have promised to release the dataset for public research. In this sense, this dataset, with sufficient diversity, fine details, and aesthetics, together with the designs in algorithm, could be valuable contributions to the community. Based on the recommendations of the reviewers, AC believes that this paper merits acceptance.